# Essentiality and dynamic expression of the human tRNA pool during viral infection

Noa Aharon-Hefetz, Michal Schwartz [ID], Einav Aharon, Noam Stern-Ginossar, Orna Dahan [ID] [✉] & Yitzhak Pilpel [ID] [✉]

## Abstract

Human viruses rely on host translation resources, including the cellular tRNA pool, because they lack tRNA genes. Using tRNA sequencing, we profiled mature tRNAs during infections with human cytomegalovirus (HCMV) and SARS-CoV-2. HCMV-induced alterations in mature tRNA levels were predominantly virus-driven, with minimal influence from the cellular immune response. Certain post-transcriptional modifications, correlated with tRNA stability, were actively manipulated by HCMV. By contrast, SARS-CoV-2 caused minimal changes in mature tRNA levels or modifications. Comparing viral codon usage with proliferation- versus differentiation-associated codon-usage signatures in human genes revealed striking divergence. HCMV genes aligned with differentiation codon usage, whereas SARS-CoV-2 genes matched proliferation codon usage. Structural and gene-expression genes in both viruses showed strong adaptation to host tRNA pools. Finally, a systematic CRISPR screen of human tRNA genes and tRNA-modifying enzymes identified specific tRNAs and enzymes that either enhanced or restricted HCMV infectivity and influenced cellular growth. Together, these data define a dynamic interplay between the host tRNA landscape and viral infection, illuminating the mechanisms governing host–virus interactions.

**Keywords** tRNA; Viral Infection; Gene Expression; Functional Genomics
**Subject Categories** Microbiology, Virology & Host Pathogen Interaction; RNA Biology

## Introduction

A major challenge for viruses upon entering a host cell is executing their gene expression programs. To establish infection, viruses must rapidly and efficiently translate large portions of their proteome while evading or countering the host's antiviral defenses. Unlike many bacteriophages (Guerrero-Bustamante and Hatfull, 2024; Limor-Waisberg et al, 2011), human viruses do not encode their own tRNA genes (Albers and Czech, 2016) and therefore depend entirely on the host's tRNA pool and associated translation factors

to meet their protein synthesis demands (Hoang et al, 2021; Rozman et al, 2023).

Host cells maintain a balance between the supply of tRNAs and the codon demand of their transcriptome, ensuring efficient and accurate translation (Boël et al, 2016; Frumkin et al, 2018; Hanson and Coller, 2018; Rudolph et al, 2016). Viruses exploit this system through strategies such as matching their codon usage to that of the host (Bahir et al, 2009; Hernandez-Alias et al, 2021) or disrupting the host's equilibrium between tRNA availability and codon demand (Pavon-Eternod et al, 2013; Van Weringh et al, 2011).

Manipulation of the tRNA pool can occur via altered transcription of tRNA genes or by modifying the chemical makeup of tRNAs (Pan, 2018). tRNAs are among the most extensively post-transcriptionally modified RNAs, with modifications occurring at specific nucleotides and positions within the molecule (Lucas et al, 2024; Suzuki, 2021). These modifications influence tRNA stability (De Zoysa et al, 2024; Wang et al, 2023), decoding accuracy (Giguère et al, 2024; Krueger et al, 2024; Saleh and Farabaugh, 2024), and other critical functions. For example, knockout of the wybutosine 'writer' TYW1 increases ribosomal frameshifting at the HIV gag-pol slippery site, potentially disrupting the gag–pol protein ratio and impairing viral assembly (Rak et al, 2021; Rosselló-Tortella et al, 2020).

Recent advances in tRNA sequencing (Behrens et al, 2021; Zheng et al, 2015) have greatly improved our ability to quantify both tRNA abundance and modification levels. In human cells, tRNA-seq has revealed major shifts in the tRNA pool during transitions between proliferative and differentiated states (Gingold et al, 2014; Rak et al, 2021; Santos et al, 2019; Zhang et al, 2018), across tissues (Pinkard et al, 2020; Zhang et al, 2018), and in response to stress (Ball et al, 2022; Ling et al, 2014). It also enables accurate monitoring of modification dynamics (Zhang et al, 2022).

However, observational profiling alone cannot reveal the functional consequences of these changes. A previous proof-of-principle CRISPR–Cas9 screen targeting 20 tRNA families demonstrated that some tRNAs are essential for cell proliferation, while others are required for cell-cycle arrest, effects that did not always align with changes in tRNA abundance (Aharon-Hefetz et al, 2020).

In this study, we combine tRNA-seq and functional genomics to investigate how two very different viruses, human cytomegalovirus (HCMV) and SARS-CoV-2, interact with the host tRNA pool. We measured changes in tRNA abundance and epitranscriptomic modifications following infection and found that HCMV induces

Department of Molecular Genetics, Weizmann Institute of Science, Rehovot 76100, Israel. ✉E-mail: orna.dahan@weizmann.ac.il; Pilpel@weizmann.ac.il

substantial alterations, whereas SARS-CoV-2 leaves the tRNA pool largely unchanged. IFN treatment revealed only minor antiviral-response effects on tRNA composition. Codon usage analysis showed that HCMV genes align with differentiation-associated codons, while SARS-CoV-2 genes align with proliferation-associated codons. Both structural and gene expression-related genes were best adapted to the infected-cell tRNA pool.

To directly test functional requirements, we designed a comprehensive CRISPR library of ~3000 sgRNAs targeting all human tRNA genes, including cytosolic, pseudogene, and mitochondrial tRNAs, as well as all known tRNA modification enzymes. Screening under normal growth and HCMV infection conditions identified specific tRNAs and modification enzymes that either promote or restrict viral replication, providing new insights into the central role of tRNAs in both cellular homeostasis and viral pathogenesis.

## Results

### Mature tRNA levels are modulated in response to HCMV infection

We investigated changes in mature tRNA expression during HCMV infection of human foreskin fibroblasts (HFFs) using tRNA sequencing of both charged and uncharged molecules. The HCMV genome replication begins ~20 h post-infection (hpi), with progeny release starting at ~4 days. Thus, samples were collected at 5, 16, 24, and 72 hpi. Experiments were done in three biological replicates. Significant changes in cytosolic tRNA abundance appeared by 24 hpi and became more pronounced at 72 hpi, with many tRNAs that were lowly expressed in uninfected cells upregulated during infection (Fig. 1A–C).

We also quantified mitochondrial tRNA levels, normalizing to the mitochondrial genome copy number (~300 per fibroblast) (Jiang et al, 2020). These levels were similar to those of the least abundant cytosolic tRNAs and showed a slight increase at 72 hpi (Appendix Fig. S1A), potentially reflecting increased mitochondrial content in infected cells (Combs et al, 2020) or altered mitochondrial tRNA gene expression.

In humans, most tRNA families contain multiple genes with identical anticodons, i.e., tRNA isodecoders, which can be differentially expressed depending on cell type and condition (Gao et al, 2024; Sagi et al, 2016). Upon HCMV infection, we observed marked differences in abundance among uniquely sequenced isodecoders within the same family (Fig. 1D), with changes in some cases spanning a range comparable to that seen between different anticodons.

To determine whether these changes were driven by the antiviral response, we treated uninfected cells with interferons (IFN), a key mediator of HCMV-induced antiviral activity (Galitska et al, 2019; Goodwin et al, 2018), and treated infected cells with ruxolitinib (ruxo), a potent IFN signaling inhibitor (Lin et al, 2009; Winkler et al, 2019). Pearson correlation analysis showed high reproducibility among replicates of each experiment (i.e., untreated, HCMV-infected with or without treatment) ($R \geq 0.85$, Appendix Fig. S1B). Principal component analysis (PCA) revealed a progressive shift in the tRNA pool during infection (Fig. 1E). IFN-treated cells showed only minor deviations from untreated cells, whereas ruxolitinib-treated and infected cells at 24 hpi

clustered between the 24 hpi and 72 hpi samples, consistent with enhanced infection upon IFN inhibition. We next examined changes at the level of tRNA families (sharing the same anticodon) by summing reads for all gene copies within a family. Hierarchical clustering of cytosolic tRNA family fold changes relative to uninfected cells (Fig. 1F) mirrored the PCA results: early infection (5 hpi) clustered with IFN-treated samples, while later time points formed a distinct group. At 5 hpi, only a few tRNAs changed (e.g., Arg-CCT), whereas later stages showed stronger alterations. Most tRNAs remained stable (e.g., Glu-TTC), but several changed dramatically, with up- or downregulation of up to fourfold within 24 hpi (e.g., Arg-CCT, Val-AAC). Given the long half-life of human tRNAs (~100 h) (Choe and Taylor, 1972) and the cell cycle arrest induced by HCMV (Bogdanow et al, 2021), neither transcriptional shutdown nor dilution by cell division can explain such rapid declines. Instead, active degradation is the most plausible mechanism. Altogether, these results suggest that IFN signaling, which constitutes the primary innate cellular anti-viral response to HCMV, has only a minor influence on tRNA pool changes, which are largely virus-driven, consistent with previous findings that HCMV upregulates tRNA transcription machinery components (Dremel et al, 2023).

Several tRNAs exhibited striking temporal dynamics: Ser-AGA decreased early, then rose more than fourfold at later times; Pro-TGG increased twofold in IFN-treated and 5 hpi samples before returning to baseline; Pro-CGG decreased ~fourfold across all conditions (Fig. 1F). Nascent transcript sequencing (Ball et al, 2022) confirmed that transcription levels of these tRNAs remained stable during infection (Appendix Fig. S1C), indicating that post-transcriptional regulation, likely mediated by viral or host factors, drives these abundance changes, suggesting active degradation.

### Specific tRNA modifications are modulated in response to HCMV infection

The tRNA pool, and thus translation efficiency, can be regulated both through changes in tRNA abundance and through alterations in chemical modifications that affect tRNA stability and function. To assess the impact of HCMV infection on tRNA modifications, we analyzed our tRNA deep-sequencing data with a focus on modification status.

In tRNA-seq, reverse transcription to cDNA is impeded or altered when the reverse transcriptase (TGIRT) encounters many of the modified nucleotides, leading to either mismatched bases or premature termination. These characteristic patterns can be used to detect and quantify changes in modification levels (Behrens et al, 2021; Kimura et al, 2020; Rajan et al, 2022; Rak et al, 2021). We found that most tRNA modifications remained unchanged between uninfected and late-infected cells (Fig. 2A, $r = 0.95$, $p < 0.001$). While tRNA sequencing enables high-throughput detection of tRNA modifications, due to its limitations in comparison to LC-MS/MS (Herbert et al, 2024), some types of tRNA modifications cannot be observed and are therefore not reported here. However, certain modifications showed significant alterations. Specifically, dihydrouridine levels were reduced in Val-CAC, Val-TAC, and Gly-CCC tRNAs at positions 20, 20, and 19, respectively (Fig. 2A). As dihydrouridine promotes tRNA stability (Faivre et al, 2021), this reduction correlates with lower expression of these tRNAs during infection (Fig. 1F). Notably, tRNAs carrying dihydrouridine

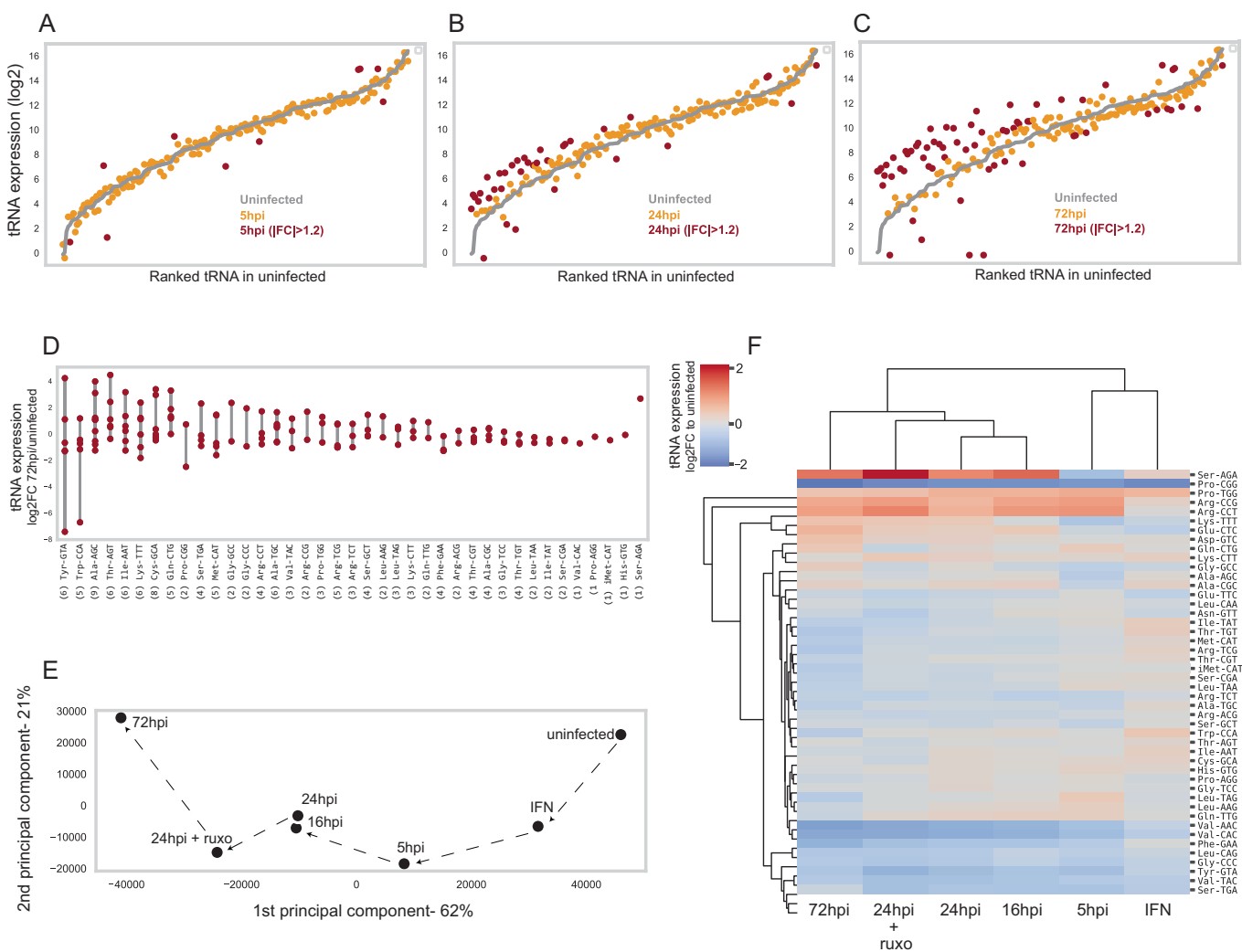

**Figure 1. Changes in the tRNA levels in HCMV-infected HFF.**

(A–C) Cytosolic tRNA levels (log2) of uninfected HFF (in gray lines) and HCMV-infected cells (in orange-red) in a time-course manner. The tRNAs are ordered based on their expression level in uninfected cells. tRNA genes that differentially expressed (|FC|> 1.2) are marked in dark red. (A) 5 h post-infection (hpi); (B) 24 hpi; (C) 72 hpi. The tRNA pool at 16 hpi is highly similar to that at 24 hpi; therefore, it is not shown here. (D) Differences in the expression level of tRNA isodecoders. The log2 fold change in expression at 72 hpi relative to uninfected cells for each tRNA isodecoder is presented for each tRNA isoacceptor. The numbers in parentheses indicate the number of isodecoders with unique sequences detected for each tRNA isoacceptor. (E) Principal component analysis (PCA) of the HCMV-infected samples based on the mean tRNA levels of three biological repeats. The dashed arrows highlight the gradual progression of the infection. (F) Hierarchical clustering of the HCMV-infected samples and tRNA families based on the averaged changes (two or three biological repeats) in the summed read fraction of each tRNA family relative to the uninfected sample (log2).

modification are among the most abundant species in uninfected cells (Appendix Fig. S2A), suggesting targeted, rather than global, modulation of this modification during infection.

We also examined modifications in the anticodon loop that influence ribosomal decoding (Rapino et al, 2017). The ms²t⁶A modification at position 37 of Lys-TTT increased in IFN-treated cells but decreased during infection (Fig. 2B; Appendix Fig. S2B). In contrast, m³C at position 32 of Ser-TGA-2-1 was mildly upregulated by IFN treatment and strongly upregulated during infection (Fig. 2C). Mitochondrial tRNAs- Ser-TGA, Trp-TCA, and Phe-GAA, displayed reduced modification levels in HCMV-infected cells (Fig. 2A).

Both ms²t⁶A and m³C have been linked to proliferative states in mammalian cells (Ignatova et al, 2020; Rak et al, 2021; Rosselló-

Tortella et al, 2020; Smith et al, 1985), consistent with the notion that HCMV infection favors a cell cycle state conducive to viral replication (Bogdanow et al, 2021; Fortunato et al, 2002).

Together, these results indicate that during HCMV infection, specific tRNA modification levels are altered, particularly highly abundant cytosolic species and modifications in the anticodon loop. The host antiviral stress response has a comparatively minor, and in some cases opposite, influence.

## HCMV genes favor differentiation-associated codons, optimized for the infected cell tRNA pool

To explore the relationship between tRNA availability and codon usage during HCMV infection, we compared the codon usage of

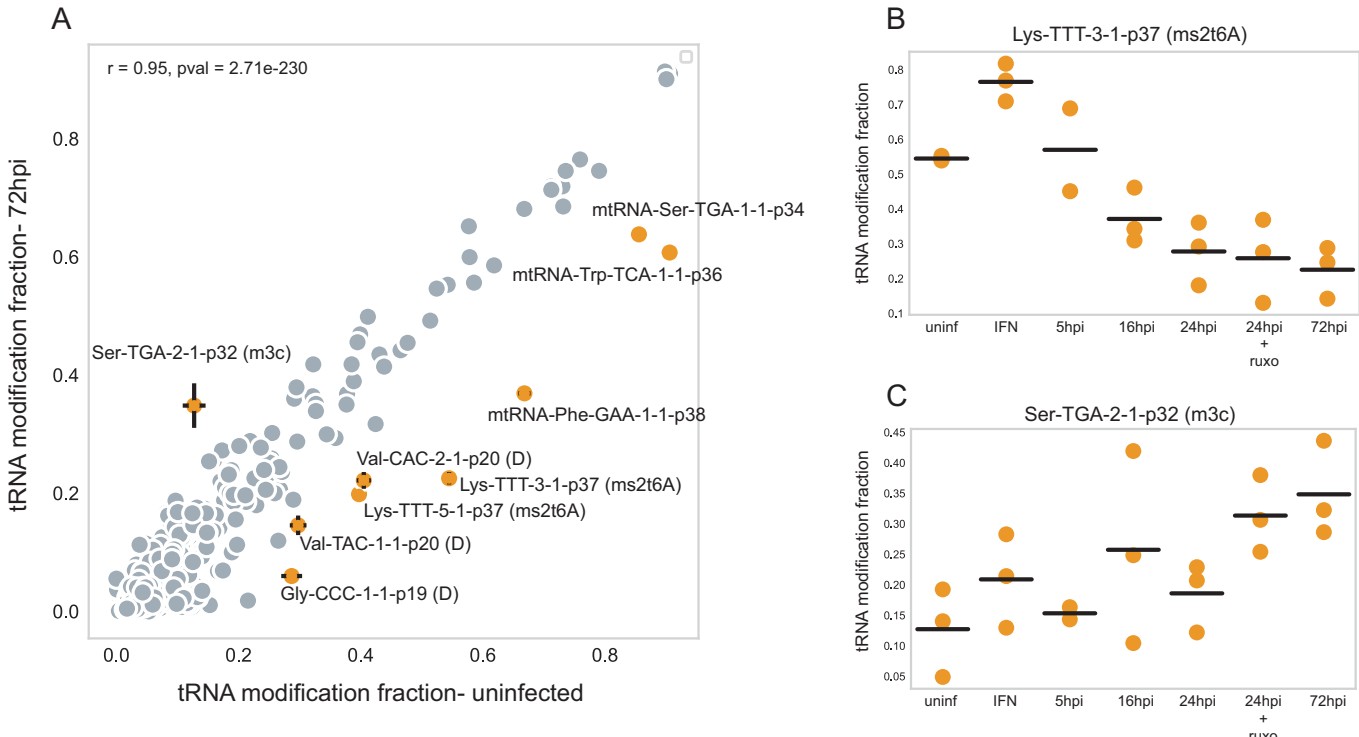

**Figure 2. Changes in the tRNA modification levels in HCMV-infected HFF.**

(A) A scatter plot describes the change in the averaged tRNA modification level in the uninfected sample (x-axis) relative to 72 hpi (y-axis). Modification levels significantly regulated following infection are marked in orange, with error bars showing the confidence interval calculated from three biological repeats. The names include the tRNA name, the position in the tRNA gene, and the type of base modification. Pearson r = 0.95, p-value = 2.71e-230. (B, C) Change in the tRNA modification level along HCMV infection on (B) Lys-TTT-3-1 gene, position 37, modification ms2t6A; (C) Ser-TGA-2-1, position 32, modification m³C. For each sample, each dot represents a biological replicate (3 replicates), and the lines represent the averaged modification level.

HCMV genes to that of human transcripts. In mammalian genes, codon usage often reflects distinct functional programs: proliferation and cell-autonomous processes versus differentiation and multicellularity (Gingold et al, 2014; Hernandez-Alias et al, 2020; Zviran et al, 2018). We quantified the similarity of each human and viral gene to these two codon usage signatures (Fig. 3A, B; Appendix Fig. S3A).

Among human transcripts (O'Leary et al, 2016), 30% strongly align with differentiation codon usage (y-axis, r > 0.8), 18% with proliferation codon usage (x-axis, r > 0.75), and ~5% show low correlation with either (r < 0.6 on both axes; Appendix Fig. S3A). For the 146 canonical HCMV ORFs, those expressed in a temporal cascade after infection (Rozman et al, 2022; Stern-Ginossar et al, 2012), 39% align with differentiation codon usage (y-axis, r > 0.8), and none align with proliferation codon usage (x-axis, r > 0.75) (Fig. 3A). This preference for differentiation codon usage is consistent with HCMV's strategy to arrest host cells at the G1/S boundary (Bogdanow et al, 2021; Fortunato et al, 2002) and may also facilitate latency and reactivation in non-dividing cells (Schwartz and Stern-Ginossar, 2023). Notably, 22% of canonical ORFs do not match either signature (r < 0.6 for both axes), a relatively large fraction given that most human-infecting viruses exhibit strong codon adaptation to their host (Bahir et al, 2009).

Half of the non-canonical HCMV ORFs identified by ribosome profiling (Stern-Ginossar et al, 2012) also show low correlation with either codon usage signature (Fig. 3B). Those that do align with differentiation codon usage tend to be much longer (>1000 bp on average) than the others (~400 bp; p < 0.001, Rank-Sum test; Appendix Fig. S3B,C). Because gene length is positively associated with evolutionary conservation (Wolf et al, 2009), we suggest that long non-canonical genes may be functionally important and translationally optimized.

We next calculated the tRNA adaptation index (tAI) (Dos Reis et al, 2004) for each canonical HCMV ORF using the tRNA abundance profile of infected cells. tAI measures how well a gene's codon usage matches the available tRNA pool, serving as a proxy for translation efficiency and mRNA stability (Forrest et al, 2020). Generally, the tAI of HCMV genes from all functional categories is higher than the averaged tAI of HCMV gene calculated using randomized tRNA expression data, as presented by the red dashed line in Fig. 3C. Functional categories differed in tAI values: genes involved in gene expression, DNA replication, and virion production had higher tAI, indicating better adaptation to the host tRNA pool than genes related to host recognition, viral tropism, and immunomodulation (Fig. 3C; Appendix Fig. S3D,E).

Finally, tAI correlated positively with similarity to differentiation codon usage and negatively with similarity to proliferation codon usage (Fig. 3D,E). As HCMV primarily infects slowly dividing cells, this may suggest that the proliferation state of the infected cells dictates the codon usage of HCMV genes.

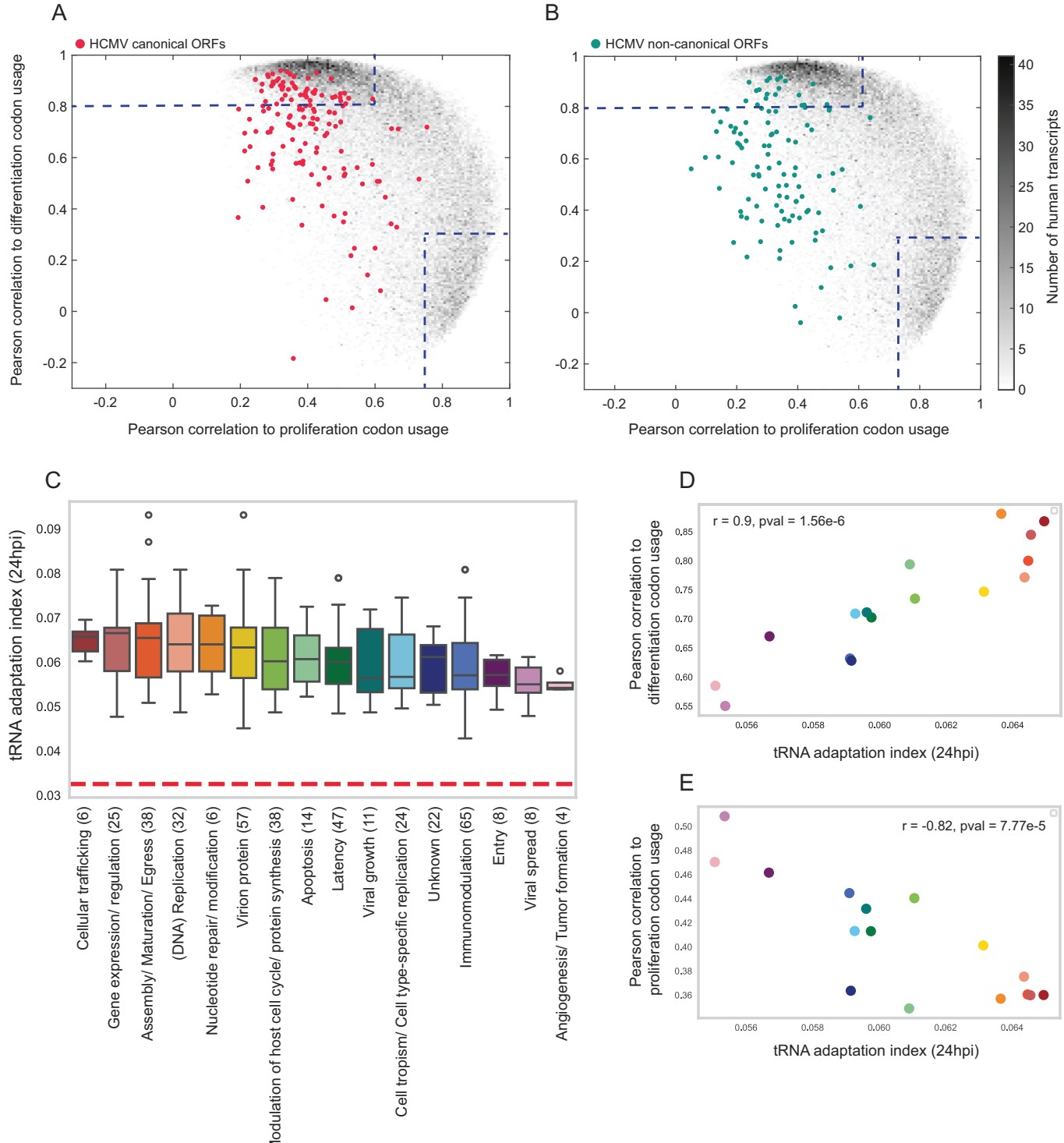

## tRNA–codon usage adaptation following SARS-CoV-2 infection

Human viruses employ diverse strategies to optimize protein synthesis. Unlike HCMV, which manipulates host translation while maintaining active protein synthesis, SARS-CoV-2 promotes viral expression through host shut-off (Finkel et al, 2021). With its rapid

replication cycle, releasing progeny by 8 hpi, we compared the tRNA pool of SARS-CoV-2-infected Calu3 cells at 6 hpi to that of uninfected cells.

The tRNA pool of infected cells was highly similar to that of uninfected cells (r = 0.94, *p* < 0.001; Fig. 4A), with only four tRNAs showing significant changes: Ile-AAT-3-1 and Trp-CCA-5-1 increased, while Ile-AAT-2-1 and Phe-GAA-2-1 decreased.

**Figure 3. Codon usage adaptation of HCMV ORFs to the tRNA pool of HCMV-infected HFF.**

(A, B) Density plots of the Pearson correlation measured for each transcript to the human proliferation (x-axis) and differentiation (y-axis) codon-usage signatures. Gray dots denote human transcripts, whereas color dots denote HCMV ORFs. The color bar (gray) represents the number of human transcripts (36,762 in total). The dashed lines represent the Pearson coefficients that determine high similarity to the proliferation or differentiation codon-usage signatures. (A) HCMV canonical ORFs (146 in total), (B) HCMV non-canonical ORFs (110 in total). The analysis did not include short non-canonical ORFs (<58 codons). (C) A box plot depicting the tRNA adaptation index (tAI) of canonical HCMV ORFs as calculated based on the tRNA pool of 24 hpi. HCMV ORFs are grouped according to their functionality, as was determined by Ye et al, 2020. The central line represents the median; the box bounds are defined by the 25th and 75th percentiles (IQR). Whiskers extend to the most extreme data points within 1.5 times the interquartile range (IQR). Points beyond the whiskers are outliers. The number of genes in each category is mentioned in brackets next to the category along the x-axis. The dashed red line represents the mean tAI level of all genes after randomizing tRNA expression among tRNA types (D, E) Scatter plots showing the averaged tAI of functional groups of HCMV ORFs at 24 hpi (x-axis) and their averaged Pearson correlation to (D) differentiation codon usage (Pearson r = 0.9, p-value = 1.56e-6), (E) proliferation codon usage (Pearson r = −0.82, p-value = 7.77e-5) (y-axis). The color code of each functional category corresponds to Fig. 3C.

Modification levels were virtually unchanged between infected and uninfected cells (r = 0.99, $p < 0.001$; Fig. 4B).

We calculated the tRNA adaptation index (tAI) for each SARS-CoV-2 gene using tRNA abundances from both conditions. Similar to HCMV genes, the tAI of SARS-CoV-2 genes is higher than the average tAI of those genes calculated using randomized tRNA expression data, as presented by the red dashed line in Fig. 4C. Structural genes (N and M proteins) displayed high tAI values, suggesting efficient translation, whereas cell entry genes (S and E proteins) had lower tAI, indicating less optimization (Fig. 4C; Appendix Fig. S4).

Codon usage analysis revealed that SARS-CoV-2 genes align strongly with the proliferation codon usage signature, but not with the differentiation signature (Fig. 4D). This is in sharp contrast to HCMV ORFs, which preferentially align with differentiation codon usage, similar to the majority of human transcripts (Fig. 3A).

These differences likely reflect the viruses' preferred host cell types and their division rates: SARS-CoV-2 tends to infect rapidly dividing cells, whereas HCMV favors quiescent or slowly dividing cells (Hernandez-Alias et al, 2021). This relationship between codon usage and host cell proliferation rate could be leveraged to predict viral tropism from codon usage patterns.

In summary, while both viruses exhibit codon usage adaptation, especially in structural and gene expression-related genes, dynamic changes in the host tRNA pool occur only during HCMV infection, but not during SARS-CoV-2 infection.

## A screen for tRNA essentiality in cellular growth reveals specific essential tRNAs and minimal impact of most tRNA modification enzymes

In our previous work, we used a CRISPR-based tRNA knockout library to study the essentiality of selected tRNA families in cell proliferation and cell-cycle arrest (Aharon-Hefetz et al, 2020). Here, we developed a comprehensive tRNA-CRISPR library covering all human tRNA genes, with multiple sgRNAs per gene, and applied it to both normal growth and HCMV infection conditions.

The library targets all tRNA types, cytosolic, pseudo, and mitochondrial, as well as tRNA modification enzymes (Fig. 5A). Each sgRNA was designed to target as many genes as possible within a tRNA family while minimizing off-target effects on other families (see Appendix Supplementary text). Five control sub-libraries were included: dependency factors essential for growth or infection, restriction factors that reduce growth or infection, and a set of non-targeting sgRNAs as negative controls. The four former

control libraries were based on a prior genome-wide CRISPR screen performed in the same cell type, which analyzed restriction and dependency factors for both cell growth and HCMV infection (Hein and Weissman, 2022). For detailed discussion regarding sgRNA library design and its targeting quality, see Appendix Supplementary text and Appendix Figs. S7 and S8.

We assessed tRNA and modification enzyme essentiality for uninfected HFF growth using a cell competition assay. HFF cells were transduced at low MOI (0.3), ensuring most cells were targeted in only one tRNA family. After antibiotic selection and recovery, we compared sgRNA representation in the initial (ancestor) and post-competition populations using MAGeCK (Li et al, 2014). Figure 5B shows the direction and magnitude of each gene's effect on growth, and a hypergeometric test quantified enrichment within sub-libraries (Fig. 5C).

Controls behaved as expected: sgRNAs targeting known growth dependency genes (e.g., PSMA5, BCL2L1) were significantly depleted, while those targeting growth-restricting genes (e.g., TP53, CDKN1A) were enriched in the post-competition cell population ($p < 1e-3$) (Fig. 5C). Among tRNAs, only the functional cytosolic group showed a significant growth impact ($p \approx 1e-7$), with many sgRNAs depleted in competing cells. Pseudo tRNAs, mitochondrial tRNAs, and tRNA modification enzymes showed no overall effect, consistent with pseudo-tRNAs being unexpressed (Appendix Fig. S5A) and CRISPR's inability to target mitochondrial genomes (Yin et al, 2022). TRMT112 was the sole essential modification enzyme, aligning with its known role in activating multiple methyltransferases (Stelzer et al, 2016).

The general lack of essentiality among modification enzymes was unexpected, given their roles in post-transcriptional RNA modification and other core processes. Essential genes are often removed from the competing population relatively quickly, a process that might result in their dropouts. To rule out a dropout scenario, we validated that sgRNAs targeting tRNA modification enzymes received sufficient read counts in the ancestor population relative to other protein-coding sub-groups (Appendix Fig. S5B). In support of the phenomenon of lack of essentiality of tRNA modification enzymes in the CRISPR screen, prior CRISPR screens in HFF cells (Hein and Weissman, 2022) similarly found most tRNA modification enzymes non-essential (Appendix Fig. S5C). One exciting explanation may be redundancy through paralogs or backup pathways (Kafri et al, 2006, 2008).

We also found substantial variation in essentiality among isodecoders within the same family (Fig. 5D). This was not explained by sgRNA efficiency scores (Appendix Fig. S5D), suggesting genuine biological differences. This aligns with previous

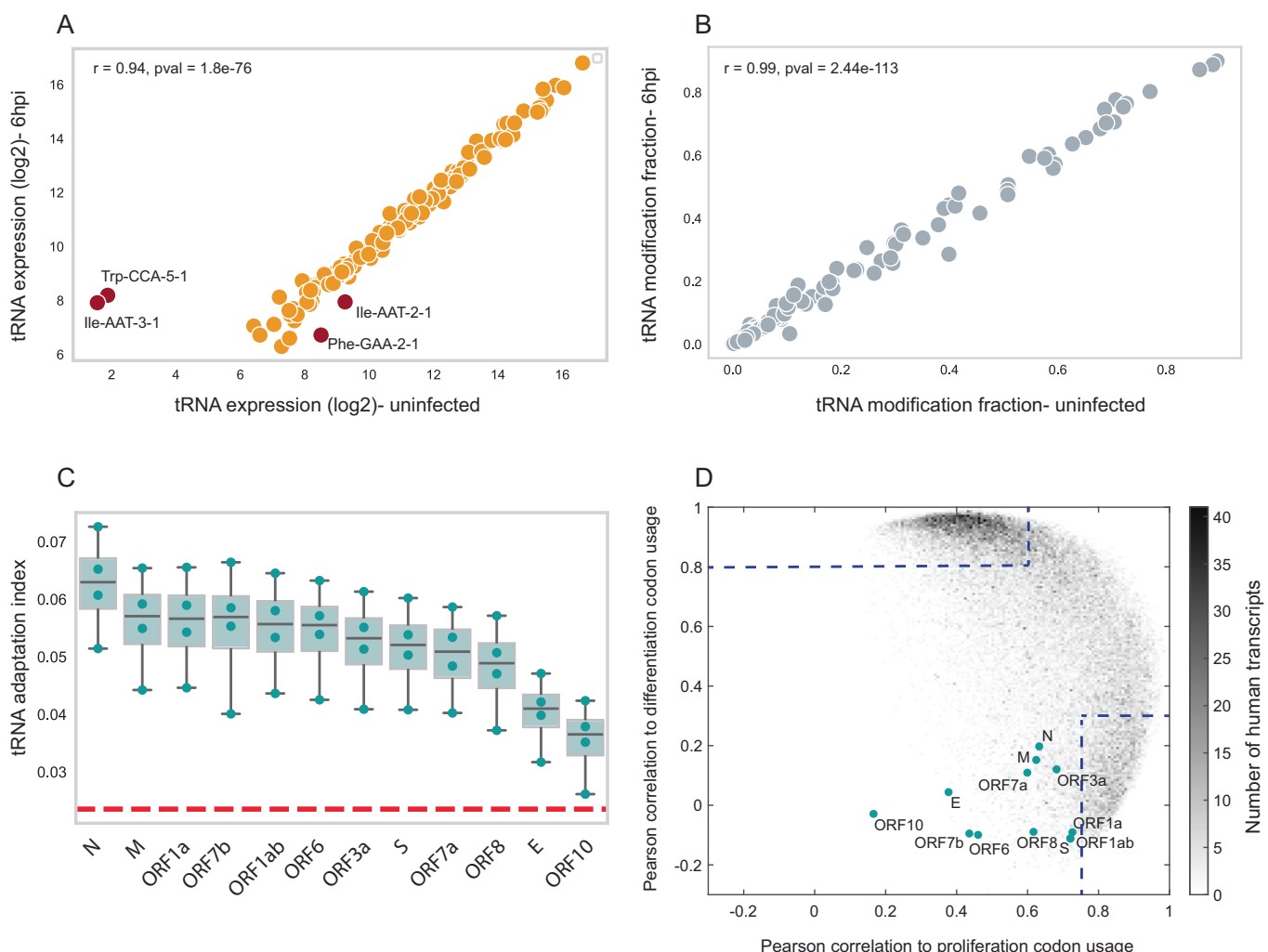

**Figure 4. Characterizing the tRNA pool and codon usage adaptation of SARS-CoV-2 genes following infection.**

(A) A comparison of the mean cytosolic tRNA level (log2) between uninfected Calu3 cells (x-axis) and 6 hpi of SARS-CoV-2 infected cells (y-axis) in two biological repeats. tRNA genes that are differentially expressed (|FC| > 1.2) are marked in dark red. Pearson r = 0.94, p-value = 1.8e-76. (B) A comparison of the mean tRNA modification fraction between uninfected Calu3 cells (x-axis) and 6 hpi of SARS-CoV-2 infected cells (y-axis), two biological repeats. Pearson r = 0.99, p-value = 2.44e-113. (C) A box plot showing the distribution of the tRNA adaptation index (tAI) values of SARS-CoV-2 genes in uninfected and infected Calu3 cells in 2 biological repeats. The central line represents the median; the box bounds are defined by the 25th and 75th percentiles (IQR). Whiskers extend to the most extreme data points within 1.5 times the interquartile range (IQR). The dashed red line represents the mean tAI level of all genes after randomizing tRNA expression among tRNA types (D) Density plot of the Pearson correlation measured for each SARS-CoV-2 transcript to the human proliferation (x-axis) and differentiation (y-axis) codon-usage signatures. Gray dots denote human transcripts, whereas color dots denote SARS-CoV-2 genes (12 genes). The color bar (gray) represents the number of human transcripts (36,762 in total). The dashed lines represent the Pearson coefficients that determine high similarity to the proliferation or differentiation codon-usage signatures.

evidence that identical tRNAs can vary in expression across cell types and conditions, and in their phenotypic impact when knocked out (Aharon-Hefetz et al, 2020; Bloom-Ackermann et al, 2014; Gao et al, 2024; Sagi et al, 2016). Finally, essentiality was moderately, negatively correlated with expression level (r = −0.41, p = 0.04; Fig. 5E), indicating that highly expressed tRNAs are generally more critical for growth.

## A CRISPR screen identifies tRNA genes and modification enzymes that modulate HCMV infection

Following our screen in uninfected cells, we next examined the essentiality of tRNA genes and tRNA modification enzymes during

HCMV infection. We employed a reporter-based CRISPR screen using an HCMV strain in which GFP is fused to the immediate early protein IE2 (IE2) (Stanton et al, 2010). HFF cells were transduced with the CRISPR library as described above, followed by antibiotic selection and recovery (Fig. 6A). Cells were then infected with IE2-GFP-labeled HCMV at a high MOI (5) and harvested after 72 h to ensure complete infection. Infected cells were sorted by flow cytometry into four groups (GFP1–4) according to GFP intensity, which reflects viral load and thus infection stage (Appendix Fig. S6A). We sequenced the sgRNA regions from each sorted population and analyzed enrichment using MAGeCK. As expected, both non-targeting controls and library-transduced cells exhibited robust infection, with the majority of cells falling into the highest

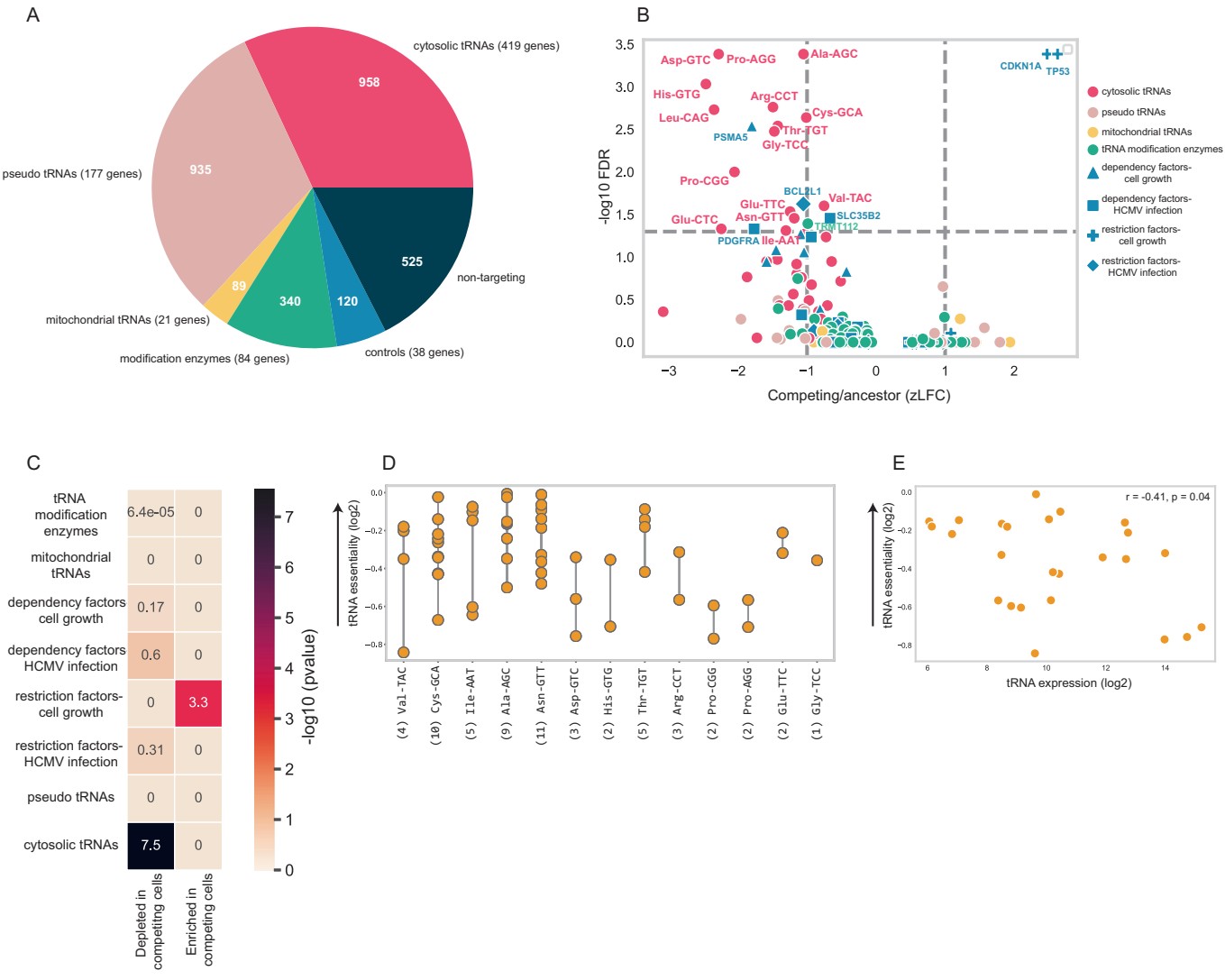

**Figure 5. Competition experiment using tRNA-CRISPR screen revealed tRNAs that are essential for HFF cell growth.**

(A) A pie chart describing the different sgRNA sub-libraries. For each sub-library, the number of targeted genes and sgRNAs is specified (the number of genes is listed in brackets, and the number of sgRNAs is indicated within the pie chart). As non-targeting sgRNAs do not correspond to existing genes, the number of genes for this sub-library is not mentioned. (B) A volcano plot showing targeted gene hits from tRNA-CRISPR screen of HFF cell growth. The x-axis shows the Z-score of log2 fold change (FC) between competing cells (3 days of competition) and ancestor samples (median of log2 FC for all high-ranked sgRNAs per gene). The y-axis shows the –log10 FDR as calculated using the MAGeCK tool. The genes are marked according to the sub-libraries. Significance is determined by FDR < 0.05 and marked with dashed gray lines. All values are calculated for three biological repeats. (C) A heat map showing the (−log10) $p$-value of the hypergeometric test, which tests the enrichment of each sub-library in one of the following groups: significantly depleted in competing cells (log2FC < 0, FDR < 0.05) or significantly enriched in competing cells (log2FC > 0, FDR < 0.05). (D) Differences in the essentiality of tRNA isodecoders that are the sole targets of their corresponding sgRNA. tRNA essentiality is determined by the median log2 FC of its corresponding sgRNA between competing and ancestor cells. The arrow points towards higher tRNA essentiality. tRNA isoacceptors are denoted on the x-axis. Each dot corresponds to a different isodecoder type within the isoacceptor family. The number of isodecoders per tRNA family is mentioned in parentheses. (E) Comparison between the (log2) expression of the tRNA isodecoder in HFF (x-axis) and their essentiality, as shown in Fig. 5D (y-axis). The arrow points towards higher tRNA essentiality. Pearson correlation r = −0.41, $p$-value = 0.04.

GFP bin (GFP4). However, library-transduced cells showed fewer GFP4 cells (70% vs. 76.7% in non-targeting sgRNA sample) and a larger fraction in GFP1–3 (28% vs. 22% in non-targeting sgRNA sample), indicating that some targeted genes are required for efficient infection (Fig. 6A, middle and lower panels).

To identify genes that, upon targeting, affect viral infection, we compared the abundance of sgRNA between lowly infected (GFP2) and highly infected (GFP4) populations. As anticipated, sgRNAs

targeting known HCMV dependency genes were enriched in GFP2 cells ($p < 0.01$), while sgRNAs targeting restriction factors were underrepresented ($p = 0.05$) (Fig. 6B,C). Notably, one known restriction factor, SEC61B, was unexpectedly enriched in GFP2 cells, consistent with earlier CRISPRi results but not with CRISPR knockout screens (Hein and Weissman, 2022), potentially explaining the discrepancy. sgRNAs targeting cell growth dependency factors were also enriched in GFP2 cells ($p = 0.001$), suggesting they

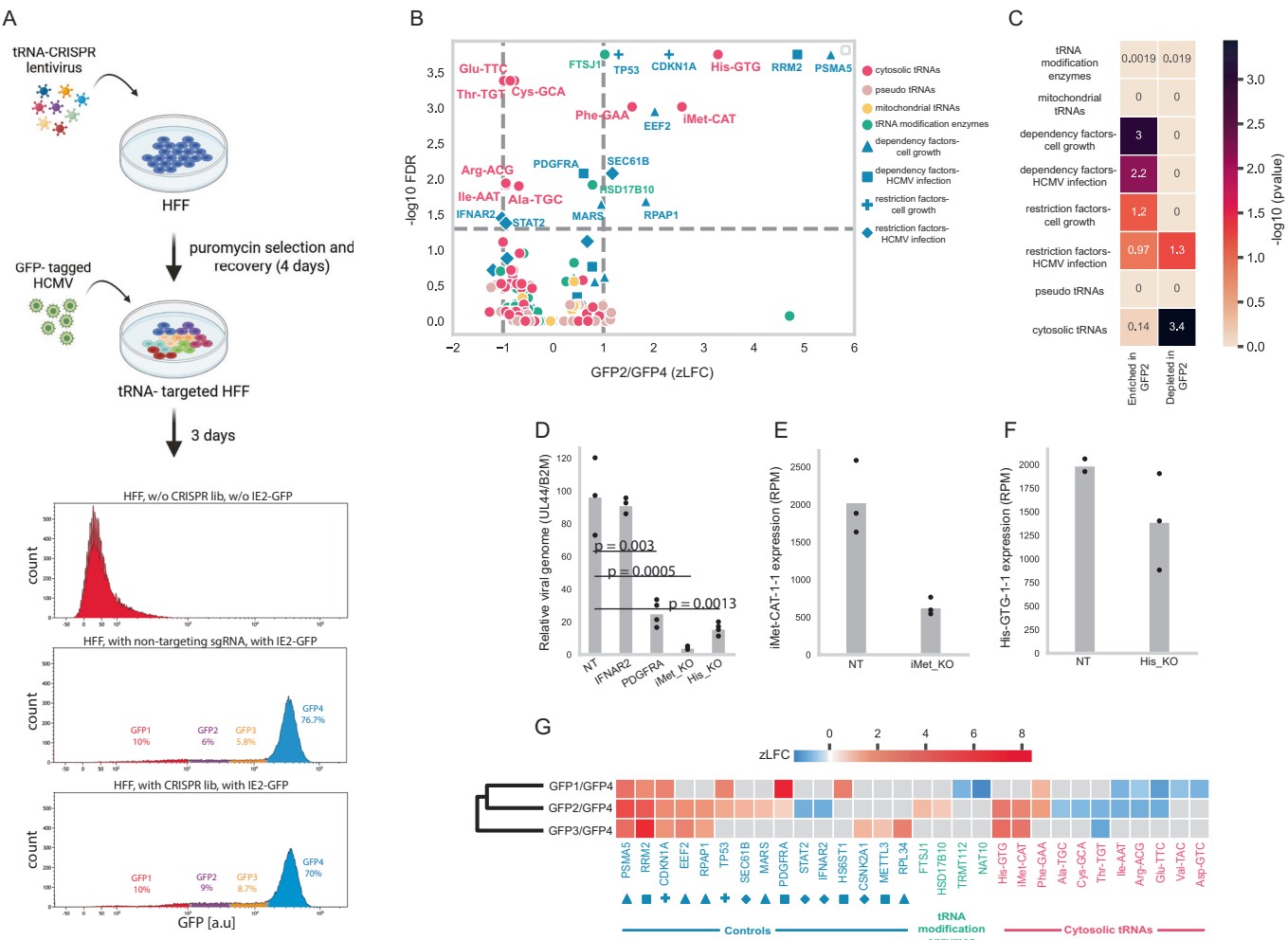

**Figure 6.** tRNA CRISPR screen in HCMV-infected HFF cells identified tRNA genes and modification enzymes that disrupt or improve HCMV infection upon CRISPR targeting.

(A) A schematic representation of the experimental setup and distributions of the measured GFP levels of uninfected HFF cells (upper panel), HCMV-infected HFF cells with non-targeting sgRNA (middle panel), and CRISPR-targeted and HCMV-infected HFF cells (lower panel). The percentage of cells in each of the GFP levels, GFP1-4, is marked in the middle and lower panels. (B) A volcano plot for targeted gene hits from tRNA-CRISPR screen in HCMV infection. The x-axis shows the Z-score of log2 FC between lowly infected cells (GFP2) and highly infected cells (GFP4). The y-axis shows the –log10 FDR as calculated from MAGeCK. The genes are marked according to the sub-libraries. Significance is determined by FDR < 0.05. All values are calculated for three biological repeats. (C) A heat map showing the (−log10) *p*-value of the hypergeometric test, which tests the enrichment of each sub-library in one of the following groups: significantly enriched in GFP2 cells (log2FC > 0, FDR < 0.05) or significantly depleted in GFP2 cells (log2FC < 0, FDR < 0.05). (D) The number of viral genomes estimated by the relative number of UL44 normalized to the B2M human gene, calculated by qPCR, in each individual CRISPR knockout. The dots in each bar depict three or four biological repeats. *P*-values represent statistical significance of differences between each group and the non-targeting control, as evaluated using Welch's t-test with Holm correction for multiple comparisons. (E, F) Expression level (RPM) determined from tRNA-sequencing of (E) iMet-CAT-1-1 gene and (F) His-GTG-1-1 gene in cells targeted by non-targeting sgRNA (NT) or by sgRNAs corresponding to the tested tRNA. The dots in each bar depict three biological repeats. (G) A heat map describes the z-score log2FC between the different lowly infected cell populations (GFP1, GFP2, GFP3) and the highly infected cells (GFP4) for the significant gene hits. Genes found as significant hits in at least one of the comparisons are presented here. Non-significant hits are marked in gray squares. The genes are colored and marked according to their sub-library, corresponding to the colors and marker shapes described in the legend of Fig. 6B. The dendrogram depicts the similarity between the comparisons based on the enrichment gene hits pattern.

also act as dependency factors for HCMV infection. Indeed, p53 levels are stabilized during HCMV infection, and the p53-p21 checkpoint is critical for regulating the host cell cycle, a process that influences HCMV replication (Bogdanow et al, 2021; Casavant et al, 2006; Fortunato et al, 2002).

Within the tRNA set, cytosolic tRNAs such as His-GTG and iMet-CAT were enriched in GFP2 cells, marking them as infection-dependency factors (Fig. 6B). Conversely, tRNAs such as Arg-ACG acted as restriction factors, with targeting reducing infectivity

(Fig. 6B). As a group, cytosolic tRNAs were significantly depleted in GFP2 cells (*p* < 0.001), whereas tRNA modification enzymes showed no overall effect. Pseudogene and mitochondrial tRNAs were similarly nonessential (Fig. 6C). Results were consistent when comparing GFP2 cells with the uninfected population (Appendix Fig. S6B).

Highly expressed tRNA isodecoders in infected cells tended to be more essential for infection, although the correlation was not significant (Appendix Fig. S6C). Importantly, GFP translation is

not biased toward codons decoded by essential tRNAs (Appendix Fig. S6D), minimizing concerns about reporter artifacts.

To validate screen hits, we performed independent individual gene knockouts of the top dependency and restriction tRNAs. Targeting two dependency tRNAs, iMet-CAT and His-GTG, was confirmed to reduce HCMV infection compared to a non-targeting control, as measured by viral genome copy number (Fig. 6D). Further, we performed tRNA-seq of the tRNA pool in these two individual gene Knockouts. The results confirmed reduced levels of the targeted tRNAs (Fig. 6E,F). Restriction tRNA knockouts, however, did not reproduce the screen phenotype. In parallel with the tRNA gene Knockouts, we performed individual CRISPR Knockouts of two protein-coding genes with opposing effects on HCMV infection, which served as controls in the screen. While PDGFRA, the cell receptor through which HCMV virions enter their host cell, reduced viral infectivity upon knockout (Fig. 6D), IFNAR2, an Interferon receptor that is part of the viral stress response, did not affect viral infectivity (relative to non-targeting sgRNA) (Fig. 6D). The low reproducibility of the restriction factor effect in CRISPR follow-up experiments was reported previously (Cross et al, 2016; Park et al, 2023). These results may reflect the context-specific nature of restriction effects and their dependence on competitive population dynamics that exist in CRISPR screens but not in individual knockout assays, highlighting the sensitivity of the CRISPR screen in identifying crucial gene factors.

To examine temporally specific essentiality, we compared sgRNA enrichment across GFP1–3 versus GFP4 (Fig. 6G). Here, we show only gene hits that are significantly enriched or depleted in at least one of the enrichment analyses. The GFP1 and GFP2 profiles were more similar to each other than to GFP3, suggesting a common role in early infection. We then compared the enrichment pattern of the sgRNAs in each GFP-sorted population to that of the known dependency and restriction factors in the library. For example, Phe-GAA was enriched in GFP1–2, mirroring PDGFRA, the fibroblast entry receptor (Soroceanu et al, 2008; Wu et al, 2018), indicating an early role in infection. In contrast, His-GTG and iMet-CAT were enriched in GFP2–3, paralleling RRM2 and METTL3, genes required for DNA replication or immune evasion, suggesting roles in later stages of infection. Several tRNAs (e.g., Glu-TTC, Arg-ACG, Thr-TGT, Ile-AAT) were depleted in GFP2 relative to GFP4, with patterns resembling those of restriction factors STAT2 and IFNAR2, suggesting that these tRNAs may hinder infection.

Among tRNA modification enzymes, sgRNAs targeting FTSJ1 and HSD17B10 reduced infectivity (enriched in GFP2), whereas TRMT112 and NAT10 knockouts increased infectivity (depleted from GFP1). Many of these enzymes act on multiple RNA substrates, making it difficult to attribute phenotypes solely to tRNA modification. FTSJ1, however, is a methyltransferase that modifies specific tRNAs in the anticodon loop at positions 32 and 34 (Safran et al, 2021). Alterations in anticodon loop modifications can impair codon–anticodon pairing and translation fidelity, potentially affecting HCMV or host protein synthesis. Enrichment of FTSJ1-targeting sgRNAs in GFP2 suggests that loss of this enzyme impairs infection, possibly by reducing modifications on tRNAs critical for translating viral or host factors required for replication.

Together, these results demonstrate that specific tRNAs are critical modulators of HCMV infection, with distinct roles at different stages of the viral life cycle, highlighting the centrality of the host tRNA pool in shaping viral replication dynamics.

## Discussion

This study examined how tRNAs and codon usage shape infection outcomes for two distinct human viruses- HCMV and SARS-CoV-2. We found that HCMV infection triggers widespread changes in the abundance of mature tRNA and specific post-transcriptional modifications, whereas SARS-CoV-2-infected cells maintain largely stable tRNA pools. Codon usage analysis revealed that HCMV genes predominantly align with differentiation-associated codons, while SARS-CoV-2 genes align with proliferation-associated codons. In both viruses, structural and gene expression-related genes showed strong adaptation to the host tRNA pool, whereas viral entry-related genes were less optimized. Using a comprehensive CRISPR library targeting all human tRNA genes and tRNA modification enzymes, we identified multiple tRNAs that either promote or restrict HCMV infection.

The stability of the tRNA pool during SARS-CoV-2 infection may reflect its short replication cycle, which ends before substantial tRNA transcriptional or processing changes can occur. In addition, SARS-CoV-2-induced host shut-off (Finkel et al, 2021) reduces cellular mRNA levels, likely lowering the demand for tRNA rebalancing. In contrast, HCMV relies on sustained host protein synthesis (McKinney et al, 2014; Tirosh et al, 2015). Our results suggest that it actively reshapes the tRNA landscape to enhance its replication.

The tRNA sequencing method we employ captures the cellular mature tRNA pool, meaning tRNAs that have been transcribed and processed. Our findings align with prior work showing that herpesviruses manipulate host tRNA transcription. For example, HSV-1 and murine gammaherpesvirus MHV68 can reprogram host tRNA transcription (Dremel et al, 2022; Tucker et al, 2020). Consistent with this, HCMV infection upregulates components of the tRNA transcription machinery and newly transcribed tRNAs (Ball et al, 2022). We extend these observations by demonstrating that HCMV modifies the mature tRNA pool not only at the transcriptional level but also via selective changes in chemical modifications. Interferon (IFN) signaling contributed minimally to these changes, reinforcing the conclusion that HCMV itself is the primary driver of the tRNA expression changes. Although specific viral regulators remain unknown, in MHV68, at least three viral proteins (ORF36, ORF45, ORF37/muSOX) induce pre-tRNA accumulation (Lari and Glaunsinger, 2023; Schaller et al, 2020; Tucker et al, 2020), suggesting HCMV may employ analogous factors.

HCMV infection altered the abundance of many tRNA isodecoders, with highly expressed variants exerting greater influence on both cell growth and viral replication. We observed reduced dihydrouridine levels in three abundant tRNAs, potentially destabilizing them (Faivre et al, 2021) and contributing to their depletion during infection. In the anticodon loop, $ms^2t^6A$ on Lys-TTT and $m^3C$ on Ser-TGA changed markedly, both modifications previously linked to proliferative states (Ignatova et al, 2020; Rak et al, 2021; Rosselló-Tortella et al, 2020; Smith et al, 1985). Since HCMV manipulates host cells to re-enter the cell cycle before arresting them at G1/S (Bogdanow et al, 2021; Fortunato et al,

2002), the level change in these modifications likely reflects a cell cycle stage optimized for viral DNA replication (Gupta and Mlcochova, 2022; Lembo et al, 1999).

By comparing viral codon usage with human proliferation/differentiation signatures, we show that HCMV's preference for differentiation codons and SARS-CoV-2's preference for proliferation codons mirrors their respective tissue tropisms (Hernandez-Alias et al, 2021). While most human-infecting viruses are well adapted to host codon usage (Charles et al, 2023), we identified a substantial subset of HCMV genes with low correlation to either signature, suggesting possible adaptation to other human gene categories, such as stress-response programs.

We used tRNA sequencing to measure changes in the mature tRNA pool, and the tRNA adaptation index (tAI) to estimate the balance between tRNA supply and demand and translation efficiency. We note that these measurements do not directly report the concentration of the aminoacyl-tRNA-EF-Tu-GTP ternary complex that delivers tRNAs to the ribosome. In particular, post-transcriptional steps, i.e., aminoacylation, EF-Tu/GTP binding, ternary-complex delivery, codon-anticodon pairing, and kinetic proofreading, are not captured by our assay. While some studies suggest that aminoacylation fractions can be relatively uniform across mature tRNAs under certain conditions (Evans et al, 2017), this effect may not hold universally. Accordingly, tAI that is calculated using sequencing-based mature tRNA pools should be viewed as reflecting the potential for decoding rather than a calibrated measure of elongation.

To assess tRNA essentiality in HCMV infection, we designed the first comprehensive CRISPR library targeting all tRNA modification enzymes and human tRNA genes, including functional cytosolic tRNAs and pseudogenes. Here, we examine the newly designed CRISPR library in the context of cellular growth and viral infection. However, this library can be applied to various cellular conditions and contexts. We previously reported on a small-scale CRISPR library in which we targeted 20 tRNA families using a single sgRNA per tRNA family (Aharon-Hefetz et al, 2020). The newly designed CRISPR library identified 15 of 49 cytosolic tRNA families as essential for HFF growth. Unlike HeLa cells, which rely more heavily on proliferation-related tRNAs (Aharon-Hefetz et al, 2020), HFFs depend on both proliferation and differentiation-associated tRNAs, consistent with their slow division rate.

We used a GFP reporter in the HCMV infection CRISPR screen to monitor the dependency of infectivity stages on tRNA families and tRNA modification enzymes. The screen revealed both dependency and restriction tRNAs. Notably, His-GTG, a tRNA previously detected inside HCMV virions and implicated in capsid stability (Liu et al, 2021), and the initiator methionine tRNA (iMet-CAT) were validated as infection-promoting factors. Unexpectedly, several tRNAs behaved as restriction factors in the pooled screen, but this signal was not recapitulated in individual perturbation tests. Possible mechanisms by which tRNAs would restrict infection are via interactions with antiviral pathways, such as tRNA cleavage by Schlafen proteins (Yang et al, 2018) or by slowing translation elongation to improve protein folding (Gorochowski et al, 2015).

Compared side by side, the proliferation screen (Fig. 5B) and the HCMV infection screen (Fig. 6B) identify largely distinct hit sets and differ in the spread of effect sizes. Notably, many hits that score strongly for infection modulation do not show commensurate effects on proliferation, implying that the HCMV phenotype is not simply a by-product of generic growth fitness. These differences are most parsimoniously explained by biological specificity, i.e., the infection screen prioritizes pathways associated with viral entry/trafficking, or innate immunity, whereas the growth screen highlights housekeeping or metabolic functions, with only a limited intersection expected between the two. Surprisingly, most tRNA modification enzymes had little impact on cell growth or HCMV infection. This could reflect technical limitations in CRISPR targeting (Horlbeck et al, 2016; Yuen et al, 2017), a limited role for these modifications under the tested conditions, or functional redundancy among modification enzymes (Kafri et al, 2006, 2008). Further investigation is needed to dissect their contributions and potential compensatory mechanisms.

In summary, our work reveals distinct viral strategies for engaging the host translation machinery. HCMV reshapes the mature tRNA pool and exploits codon usage matching to maximize translation efficiency, while SARS-CoV-2 relies on codon adaptation without altering host tRNA abundance. These insights highlight tRNAs as active participants in virus-host interactions, suggesting new targets for antiviral intervention.

# Methods

**Reagents and tools table**

| Reagent/Resource | Reference or Source | Identifier or Catalog Number |
|---|---|---|
| **Experimental models** | | |
| HEK293T cell (*H. sapiens*) | ATCC | CRL-3216 |
| Human Foreskin Fibroblasts (*H. sapiens*) | ATCC | CRL-1634 |
| Calu3 cells (*H. sapiens*) | ATCC | HTB-55 |
| HCMV Merlin strain with a GFP tag fused to the IE2 protein | Stanton et al, 2010 | N/A |
| NEB stable *E. coli* strain | NEB | C3040H |
| competent stbl3 bacterial strain | Invitrogen | C7373-03 |
| **Recombinant DNA** | | |
| LentiCRISPR v2 | Addgen | 52961 |
| PMD2.G | Addgen | 12259 |
| psPAX2 | Addgen | 12260 |
| **Oligonucleotides and other sequence-based reagents** | | |
| sgRNA oligo pool library | This study | Dataset EV2 |
| qPCR primers | This study | Table EV1 |
| PCR mega-primers | This study | Table EV2 |
| **Chemicals, Enzymes and other reagents** | | |
| DMEM high glucose | Biological Industries | 01-052-1A |
| RPMI | Thermo Fisher | **11875093** |
| Fetal bovine serum (FBS) | Life Technologies | **26-140-095** |
| Interferon α (IFNα) | PBL | 11200-2 |
| Interferon β (IFNβ) | Peprotech | 300-02BC-20 |
| ruxolitinib | InvivoGen | 941678-49-5 |
| jetPEI transfection reagent | Polyplus | 101-10N |

| Reagent/Resource | Reference or Source | Identifier or Catalog Number |
|---|---|---|
| Hexadimethrine Bromide | Sigma Aldrich | H9268 |
| puromycin | Thermo Fisher | A1113802 |
| Tri-Reagent | Sigma-Aldrich | T9424 |
| TGIRT™-III Enzyme | InGex | LLC |
| ESP3I enzyme | Thermo Fisher | FD0454 |
| BsmB1-V2 restriction enzyme | NEB | R0580 |
| T7 ligase | NEB | M0202 |
| QuickCIP | NEB | M0525S |
| Tango buffer | Thermo Fisher | BY5 |
| rCutSmart buffer | NEB | B6004S |
| Kapa HiFi ready mix | Roche | KK2602 |
| SYBR Green PCR master-mix | Applied Biosystems | 4309155 |
| NEBuilder HiFi DNA Assembly Master Mix | NEB | E2621S |
| ampicillin | Merck | A1593 |
| DTT | Sigma-Aldrich | D9779 |
| **Software** | | |
| CRISPR sgRNA design tool of Benchling | https://benchling.com | |
| Blast tool | Altschul et al, 1990 | |
| CRISPR clue tool | Becker et al, 2020 | |
| BD FACS Diva software v8.0.1 | BD Biosciences | |
| MAGeCK software | Li et al, 2014 | |
| **Other** | | |
| Illumina NovaSeq 1.5B | Illumina | |
| PCR clean-up kit | Promega | A9281 |
| NucleoBond Xtra Midi kit | Macherey-Nagel | 740412.50 |
| Monarch Spin High-Capacity DNA cleanup kit | NEB | T1135S |
| Wizard Plus SV Minipreps DNA Purification kit | Promega | **A1330** |
| BD FACSAria Fusion instrument | BD Immunocytometry Systems | |
| QuantStudio 12 K Flex | Applied Biosystems | |

## Methods and protocols

### Cell culture

HEK293T cells (ATCC) and Human Foreskin Fibroblasts (ATCC) were grown in DMEM high glucose medium (Biological Industries) supplemented with 10% heat-inactivated FBS (Life Technologies), 1% penicillin/streptomycin (P/S), and 1% L-Glutamine.

Calu3 cells (ATCC) were cultured in 6-well or 10-cm plates with RPMI (Thermo Fisher) supplemented with 10% fetal bovine serum (FBS, Life Technologies), MEM non-essential amino acids, 2 mM L-glutamine, 100 units per ml penicillin, and 1% Na-pyruvate.

### Mature tRNA sequencing of virus-infected cells

HFF cells (passage 27) were grown to full confluence in 10 cm plates and then infected with the HCMV merlin UL32-GFP strain (Stanton et al, 2010) at a Multiplicity of infection (MOI) of 5. Cells were incubated with the virus for 1 h, washed, and supplemented with fresh medium. Cells were harvested at 4 time points following infection- 5, 16, 24, and 72 h post-infection. HFF treated with IFN were incubated with 550 U/ml IFNα (PBL) and 700 U/mL IFNβ (Peprotech) and harvested after 5 h. Infected cells treated with ruxolitinib were supplemented with a medium containing 4 μM ruxolitinib (InvivoGen) after 1-h incubation with HCMV and harvested 24 h post-infection.

SARS-CoV-2 infection of Calu3 cells and RNA extraction were done as described in Finkel et al (2021). In short, Calu3 cells were infected with the SARS-CoV-2 virus at an MOI of 0.2 using RPMI medium without FBS for 1 h.

All cell lines and samples were harvested with Tri-Reagent (Sigma-Aldrich). Total RNA was extracted according to the manufacturer's protocol.

Library preparation for sequencing and data analysis were done as described in Rak et al (2021). In short, library preparation includes size selection of the small (<200 bp) RNA from the total RNA, deacetylation to remove the loaded amino-acid, then reverse transcription using TGIRT™-III Enzyme (InGex, LLC) and DNA-RNA hybrid primers (to capture the CCA tail of tRNA molecules). The cDNA was then PCR-amplified and pooled for deep sequencing using NovaSeq with a 75-read length. Sequencing analysis involved read trimming, alignment to mature tRNA gene sequences, followed by read count summary and variant calling to detect mutations and Indels.

### Codon usage analysis

First, whole-genome sequences of human-infecting viruses were downloaded from the Virus-Host database, updated as of November 2017 (Mihara et al, 2016). Only gene sequences that started with an AUG start codon, ended with one of the stop codons, and kept a proper reading frame were included in the following analysis. To compare the codon usage of viral genes to the proliferation-differentiation codon usage signatures, the signature was determined. The proliferation codon usage signature is the averaged codon usage of all genes included in the GO category "M phase of mitotic cell cycle" (GO:0000087). The differentiation codon usage signature is the averaged codon usage of all genes included in the GO category "pattern specification process" (GO:0007389). Codon usage was calculated for all human and viral transcripts by counting each codon and normalizing it to the total number of codons in the transcript. The codon usage signatures that were used for the analysis are described in Dataset EV1. Then, Pearson's correlation was calculated between the codon usage of each viral gene and each codon usage signature (proliferation/differentiation).

### tAI estimation from tRNA levels

Estimating tAI from viral-infected cells was done as described in Rak et al (2021). In short, tRNA availability was defined for each tRNA type (anti-codon) by the sum of reads aligned to its tRNA genes. Then, the translation efficiency of an individual codon was determined by the availability of tRNAs that serve in translating it, incorporating both the fully matched tRNA and tRNAs that contribute to translation through wobble rules (W-value).

For randomized tAI, tRNA expression profiles were randomly reassigned among tRNA types while preserving replicate structure. For each shuffled dataset, the tAI was calculated for every viral gene in each replicate and then averaged across replicates. This shuffling procedure was repeated 100 times, and the mean tAI across all genes was recorded for each round. We report the mean tAI of all randomization rounds.

### sgRNA library design

The sequences of all human tRNA genes, including functional cytosolic, pseudo, and mitochondrial tRNAs, were downloaded from the tRNA database (Chan and Lowe, 2016). The CRISPR sgRNA design tool in Benchling ([Biology Software]. (2022), retrieved from https://benchling.com) was used for sgRNA design, utilizing the default parameters. To remove sgRNAs with tRNA off-targets (sgRNAs that have high sequence similarity to tRNAs that are not part of the targeted tRNA family), we used the Blast tool (Altschul et al, 1990) and compared each sgRNA sequence (with a pam sequence NGG downstream) to the entire tRNA gene sequences (both functional cytosolic and pseudo tRNAs). sgRNAs with 0-1 mismatches relative to a non-targeted tRNA were removed from the final list of sgRNAs, provided the targeted tRNAs have at least two valid sgRNAs. For detailed discussion regarding sgRNA library design, see Appendix Supplementary text and Appendix Figs. S7 and S8.

### sgRNA library cloning

The sgRNA library was synthesized as a DNA oligo pool by Twist Biosciences. In addition to the sgRNA sequence, all oligos contained flanking regions that included several elements, such as restriction sites for cloning and primers for sub-library amplification. Oligos were designed using the CRISPR clue tool (Becker et al, 2020). The final oligo design is shown in Appendix Fig. S7H, and its sequences are listed in Dataset EV2.

The synthesized oligos were amplified by PCR using 2.5 ng of oligos, 10 μM of general primers, and 2X Kapa HiFi Ready Mix [Roche], with a total volume of 50 μL. The Tm of the PCR annealing step was 55 °C, and 14 cycles were performed to amplify the sgRNAs. The PCR product was cleaned using a PCR clean-up kit (Promega). The PCR product was then cloned into a lentiviral vector using Golden Gate cloning (Benchling, [Biology Software]. (2022), retrieved from https://benchling.com). Briefly, 50 ng of LentiCRISPR v2 (addgen) and 5 ng of PCR-amplified sgRNA pool were mixed with 1 μL of BsmB1-V2 restriction enzyme (NEB) and 1 μL of T7 ligase (NEB). Digestion and ligation of the vector to the oligos were conducted using 50 cycles of temperature changes (42.5 °C to 16 °C, each for 5 min). The mixture was heated to 60 °C for 5 min to terminate the digestion reaction. Then, the cloned vector was cleaned using standard ethanol precipitation (Sambrook and Russell, 2001) and electroporated into 50 μl of NEB stable *E. coli* strain (NEB) using 1 μl of cloning material (Applied volts—2200 V, Resistance—200, Capacitance—25 μF). Transformed cells were recovered in SOC media (0.5% yeast extract, 2% tryptone, 10 mM NaCl, 2.5 mM KCl, 20 mM MgSO₄, 0.4% glucose, pH 7.5) for an hour at 37 °C, then selected by growing on LB medium supplemented with 100 μg/ml ampicillin (Merck) for 16 h at 25 °C. Transformation yielded X800 library coverage, as determined by cell seeding on selection plates in serial dilution. Finally, cells were harvested at OD = 1 for plasmid extraction using the NucleoBond

Xtra Midi kit (Macherey-Nagel). The composition of the final library was verified by deep sequencing.

### Cell transduction with lentiviral- tRNA- CRISPR library and HCMV infection

To produce lentiviruses, HEK293T cells were seeded onto 10 cm plates to achieve an approximate cell confluence of 70% the following day. The day after, 2.5 μg of PMD2.G (Addgene) and 2.5 μg psPAX2 (Addgene) packaging vectors were co-transfected with 5 μg of the sgRNA library using 30 μl of jetPEI (Polyplus) in DMEM high glucose medium (10 ml). For the non-targeting sgRNA cell population, the transformation was done with a lentiCRISPR-V2 plasmid cloned with a non-targeting sgRNA (5'-TTTCGTGCCGATGTAACAC-3'). After 60 h, the lentivirus-containing medium was collected and centrifuged for 15 min at 3200 × g, 4 °C. The supernatant was collected in a new tube and filtered through a 0.4 μm filter. The lentivirus-containing medium was stored in aliquots at −80 °C. The cell death curve of HFF was used to determine the viral titer.

HFF (passage 25) were transduced at a low MOI (0.3) to ensure that most cells receive only one viral construct with a high probability. Media containing lentiviruses was supplemented with 8 μg/ml of Hexadimethrine Bromide (Sigma-Aldrich). 48 h after transduction, 1.75 μg/ml puromycin (Thermo Fisher) was added to select for infected cells. Then, the media was refreshed (without antibiotics), and cells were recovered for 48 h. Six days after lentiviral transduction, the cells were divided into three populations. The first was harvested immediately (i.e., ancestor samples). The second population continued to grow for 72 h in a competition mode and was harvested. This population was used to determine the effects of tRNA knockouts on cell growth. To determine the tRNA effect on HCMV infection, the third population was infected with the HCMV Merlin strain with a GFP tag fused to the IE2 protein (Stanton et al, 2010), with a high MOI of 5, as follows: cells were incubated with the virus for 1 h, washed, and supplemented with fresh DMEM medium. 72 h post-HCMV infection, cells were harvested in a medium enriched with 2% FCS, and FACS sorted into 4 cell populations based on GFP intensity (GFP1 to 4). Flow cytometry analysis was performed on a BD FACSAria Fusion instrument (BD Immunocytometry Systems), equipped with 488-, 405-, 561-, and 640-nm lasers, using a 100-m nozzle and controlled by BD FACS Diva software v8.0.1 (BD Biosciences). GFP was detected by excitation at 488 nm, and emission was collected using 502 nm longpass (LP) and 530/30 nm bandpass (BP) filters. Each subpopulation was centrifuged after sorting, and cell pellets were stored at −80 °C.

### sgRNA sequencing—library preparation and data processing

Genomic DNA was extracted from each sorted population, ancestor samples, and samples without HCMV infection using the NK lysis protocol (Chen et al, 2015). Genomic DNA was used as a template for PCR to amplify the sgRNAs, as described in Aharon-Hefetz et al (2020). Here, we used 1 μg of genomic DNA for the first PCR reaction and ran the PCR program for 16 cycles. Shifted primers were used to increase library complexity. MAGeCK software quantifies and tests for sgRNA enrichment (Li et al, 2014). The abundance of sgRNAs was first determined using the MAGeCK "count" module for the raw fastq files. To test for robust sgRNA and gene-level enrichment, the MAGeCK "test" module was used

with default parameters and the "alphamedian" method to calculate gene log fold changes.

### Quantitative real-time PCR analysis

Real-time PCR was performed on the extracted genomic DNA of the GFP-sorted cell populations using the SYBR Green PCR master mix (ABI) on the QuantStudio 12 K Flex (ABI) with specific primers (forward and reverse) amplifying UL55 (HCMV gene) and B2M (human gene). Primer sequences are shown in Table EV1. To estimate the number of viral genomes in each GFP-sorted sample, the following formula was used: *num of viral genomes* $= 2^{Ct[g]-Ct[n]}$; Ct[g] is the Ct value of the UL55 gene, and Ct[n] is the mean Ct value of all technical repeats of the normalizer gene B2M. Three technical replicates of one representative experiment are shown. Due to the low amount of genomic material of GFP1, qPCR analysis was not performed on this sorted cell population.

### Individual CRISPR knockout assay

To validate gene hits from the HCMV infection CRISPR screen, we cloned the best-performing sgRNAs from our CRISPR library into a LentiCRISPR v2 (addgen). Each sgRNA was synthesized as a mega-primer in the context of flanking regions aligned to the vector. Mega primers were synthesized by Sigma-Aldrich, and their sequences are shown in Table EV2.

For cloning, the vector was first linearized using the ESP3I enzyme [Thermo Fisher], 10X Tango buffer (Thermo Fisher), 20 mM DTT (Sigma-Aldrich), and 700 ng vector. The reaction was incubated for 1 h at 37 °C and then for 20 min at 65 °C to facilitate heat inactivation. After vector clean-up using the Monarch Spin High-Capacity DNA cleanup kit (NEB), the linearized vector was dephosphorylated using QuickCIP (NEB) and 10X rCutSmart buffer (NEB). The reaction was incubated for 10 min at 37 °C and then for 2 min at 80 °C to inactivate the heat. Then, using Gibson Assembly, the mega-primers were cloned into the linearized and dephosphorylated vector by mixing the vector with 0.2 μM of both Forward and Reverse oligos of the same sgRNA and 10 μL NEBuilder HiFi DNA Assembly Master Mix (NEB), for a total volume of 20 μL. The reaction was incubated for 1 h at 50 °C. Then, the cloned vectors were transformed into the competent STBL3 bacterial strain (Invitrogen) using a standard heat-shock transformation (Sambrook and Russell, 2001). Cloned vectors were purified from transformed colonies using a Wizard Plus SV Minipreps DNA Purification kit (Promega). Sequence validation was performed using whole-plasmid sequencing by Plasmidsaurus, which utilized Oxford Nanopore Technology with custom analysis and annotation.

After cloning, all steps including lentivirus preparation, HFF cell transduction and IE2-GFP merlin strain infection was done similarly to the CRISPR screen, as described in "Cell transduction with lentiviral-tRNA-CRISPR library and HCMV infection", except that HFF cells were plated in a 6-well plate format, and lentivirus transduction of the sgRNA-cloned CRISPR-V2 was done in MOI = 1.

Total RNA extraction and tRNA sequencing were done as described in "Mature tRNA sequencing of virus-infected cells".

Human genomic DNA and viral DNA were extracted using the NK lysis protocol (Chen et al, 2015). qPCR was performed on viral and human genes from infected cells as described in "Quantitative real-time PCR analysis". Here, UL44 gene's specific primers were used to estimate viral genome copy. Primer sequences are shown in Table EV1.

## Data availability

HCMV infection tRNA sequencing data, SARS-CoV-2 infection tRNA sequencing data, CRISPR library sgRNA sequencing data and Individual knockout tRNA sequencing data are deposited at Gene Expression Omnibus (accession no. GSE308057).

The source data of this paper are collected in the following database record: biostudies:S-SCDT-10_1038-S44320-025-00181-7.

## Peer review information

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

## Acknowledgements

We thank the European Research Council (ERC 616622), the research grant from the Sharon Zuckerman Laboratory for Research in Systems Biology and Minerva Center for Live Emulation of Evolution in the Lab for grant support. We thank Dr. Roni Rak from the Volcanic Institute for the help with tRNA sequencing and analysis. We thank Dr. Yaara Finkel, Aharon Nachshon, Batsheva Frankel Rozman, Avraham Gluck, Tal Fisher, and Tamar Arazi from Prof. Noam Stern-Ginossar's group at the Weizmann Institute of Science for their help with HCMV and SARS-CoV-2 infection and analysis. We thank the scientific and professional staff at the Weizmann Institute's life science core facilities, especially Dr. Tomer-Meir Salame from the FACS unit, Dr. Shifra Ben-Dor from the Bioinformatics unit, and Dr. Yoav Peleg from the DNA manipulation unit. Special thanks to Prof. Richard Stanton from Cardiff University for the kind contributions of HCMV strains.

## Author contributions

**Noa Aharon-Hefetz**: Conceptualization; Data curation; Formal analysis; Validation; Investigation; Visualization; Methodology; Writing—original draft. **Michal Schwartz**: Conceptualization; Supervision; Writing—review and editing. **Einav Aharon**: Data curation; Methodology. **Noam Stern-Ginossar**: Conceptualization; Supervision; Writing—review and editing. **Orna Dahan**: Conceptualization; Supervision; Writing—review and editing. **Yitzhak Pilpel**: Conceptualization; Supervision; Funding acquisition; Writing—review and editing.

Source data underlying figure panels in this paper may have individual authorship assigned. Where available, figure panel/source data authorship is listed in the following database record: biostudies:S-SCDT-10_1038-S44320-025-00181-7.

## Disclosure and competing interests statement
The authors declare no competing interests.

