## [Peer Review File · Molecular Systems Biology]

Essentiality and dynamic expression of the human tRNA pool during viral infection

Noa Aharon-Hefetz, Michal Schwartz, Einav Aharon, Noam Stern-Ginossar, Orna Dahan, and Yitzhak Pilpel

Corresponding author(s): Yitzhak Pilpel (pilpel@weizmann.ac.il), Orna Dahan (orna.dahan@weizmann.ac.il)

Review Timeline:

Transfer Date:	5th Sep 24
Editorial Decision:	9th Sep 24
Revision Received:	19th Sep 25
Editorial Decision:	22nd Oct 25
Revision Received:	3rd Dec 25
Accepted:	8th Dec 25

Editor: Jingyi Hou

Transaction Report: This manuscript was transferred to Molecular Systems Biology following peer review at Review Commons.

**Review
COMMONS**

Review #1

1. Evidence, reproducibility and clarity:

Evidence, reproducibility and clarity (Required)

I have mixed feelings regarding this manuscript. On the one hand, the authors did an impressive amount of work. On the other hand, the manuscript seems overly descriptive (writing should be more concise) without a clear message or hypothesis that is cohesive to all the presented evidence. Below, I will outline my concerns.

Regarding the first part.

1. I am not an expert in the field of viral biology and immunology. I wonder how well the IFN treatment mimics the cellular response to infection (yet without the virus). Also, how good is ruxolitinib at blocking the IFN response? I would appreciate it if you could explain both with one or two sentences and provide the necessary references.
2. (MAJOR) Can these two treatments really allow the effects of host response and viral infection to be separated? OR in other words, are these two effects really orthogonal? In my opinion, they are NOT. Fig. 1E seems to support my opinion, as the changes seen for the "IFN" sample relative to the "uninfected" sample (referred to as "changes-A" below), are parallel to the changes seen for the "24hpi + ruxo" sample relative to the "24hpi" sample ("changes-B"). More specifically, changes-A represent the host response, as argued by the author, whereas changes-B represent the elimination of the host response (due to ruxo, conditioned on the virus-driven effect). If the virus-driven effect and the host response could really be separated, one would expect changes-A and changes-B are more or less opposite. However, they appeared to be parallel, suggesting that uninfected versus infected conditions can have totally different (even opposite) host responses. More importantly, if one cannot separate the host response from virus-driven effects, the conclusion of "tRNA changes are driven by virus, not host response" is then unfounded.3. Even if we let go of this previous point and accept that these results indeed offer some support for the notion that the virus-driven effect are the main contributor to the shifts in tRNA pool, the support is at best moderate. A big gap here is "how?" I suggest the authors should at least give some insight on how virus can do that in Discussion (and mention it with one sentence in Results).
4. The authors compared the HCMV codon usage to the proliferation and differentiation signatures of human cells. But these two signatures are not compared with measured tRNA expression. It might shed some light on the general characteristics of tRNA pool shifts due to infection (towards a proliferation-like or differentiation-like signature). This fits in the

general topic of virus-host interaction and might give more evidence for the point that HCMV is adapted to a differentiation signature (as it drives the host into that state).

5. How is the dashed box in Fig3A/B chosen?

6. The tAI values shown in Fig3C-E are extremely low (compared to other reports I am aware of). Does this mean that the adaptation of viral codon usage to human cell supply is actually very weak? This is in opposition to the major claims made in this section.

7. I believe that the part about SARS-CoV-2 could be made more concise. It is sufficient to mention that results may differ from those obtained with HCMV in one paragraph.

8. Line 299 on page 11 - I do not believe codon usage between different viruses can be directly compared, let alone reaching such a conclusion. Some viruses have low CAI or tAI to humans, but they have co-evolved with humans for a long time. Furthermore, there are viruses that infect multiple hosts, but their CAI for a host with which they have long co-evolved is higher while their CAI for a host that is relatively new is lower.

9. (MAJOR) A more general comment is that there is a difference between tRNA expression and the abundance of translation-ready tRNA. The process of charging tRNA with amino acids may take a long time. It is the abundance of the charged-tRNA (the ternary complex of aminoacylated tRNA and EF-Tu-GTP) that is of biological importance. In this regard, the use of tRNA expression falls short.

Regarding the second part,

1. (MAJOR) Prior to the actual competition assay in the first high-throughput screen (cell competition assay), the authors applied two days of antibiotic selection and two days of recovery. This could result in a serious problem of false negatives or drop outs. Specifically, an sgRNA targeting an essential gene with high efficiency would kill the cells, leaving no (or a small number of) cells in the ancestor population at the beginning of the competition process. A sgRNA's enrichment in competing populations cannot be reliably estimated in such situations. I am not certain that the FDR used in Figure 5B is sufficient to address this issue. Please clarify whether it could. Providing raw counts for competing and ancestor populations would also be helpful.

2. It is also highly questionable to me the nearly negligible effects of tRNA modification enzymes. This may be explained by the point above. Indeed, the dots of tRNA modification enzymes in general appear to have higher FDR (lower y values) when compared to red dots with similar enrichment levels.

3. The screen based on IE2-GFP labeled HCMV measures a phenotype that is very difficult to interpret. Particularly, I am not sure if GFP2 and GFP3 are good controls for comparing GFP4 (GFP1 might be better). Various factors can affect GFP levels, including, but not limited to, dilution caused by a rapidly dividing host cell, unhealthy translational machinery

resulting from infection or microenvironment. My point is supported by some observations in Fig6B. For example, SEC61B, a restriction factor for HCMV infection, is enriched in the GFP2 group, contrary to expectations. It is necessary for the authors to prove with firm evidence that their choice of GFP signal thresholds is appropriate.

4. I would appreciate more information regarding why restriction factors of cell growth have a high GFP2/GFP4. Intuitively, a KO of restriction factors of cell growth should result in better growth and higher GFP, thus leading to enrichment in GFP4, not GFP2.

5. Line 404 "nonetheless"

2. Significance:

Significance (Required)

The relation between human tRNA supply and viral translation is a topic of profound biological and biomedical importance. In this study, the authors used HCMV infection as the primary model to investigate this question. Results fall into two major parts: (i) changes in the tRNA pool during viral infection, and (ii) the impact of tRNA-related gene KO on viral infection.

3. How much time do you estimate the authors will need to complete the suggested revisions:

Estimated time to Complete Revisions (Required)

(Decision Recommendation)

Between 3 and 6 months

Yes

Review #2

1. Evidence, reproducibility and clarity:

Evidence, reproducibility and clarity (Required)

In this study by Aharon-Hefetz et al., the researchers examined changes in tRNA pools during virus infections. The translation machinery plays a crucial role in virus replication. Consequently, host cells have developed sensors and effectors within this compartment to counteract viral mechanisms. The translation apparatus serves as a pivotal point in the virus-host conflict. Therefore, investigating alterations in the translation machinery during infections is vital for gaining a comprehensive understanding of the infection process.

This study offers a thorough and high-quality analysis of data in a relevant cell culture system involving two different viruses. By conducting tRNA sequencing, the researchers studied the human tRNA pool following infections with human Cytomegalovirus (HCMV) and SARS-CoV-2. Changes in tRNA expression induced by HCMV were mainly driven by the virus infection itself, with minimal impact from the cellular immune response. Interestingly, specific tRNA post-transcriptional modifications seemed to influence stability and were subject to manipulation by HCMV. Conversely, SARS-CoV-2 did not lead to significant alterations in tRNA expression or post-transcriptional modifications.

Moreover, a systematic CRISPR screen targeting human tRNA genes and modification enzymes allowed the identification of specific tRNAs and enzymes that either enhanced or reduced HCMV infectivity and cellular growth. This information enabled them to control the development of HCMV-specific tRNA modifications, highlighting the importance of these tRNA epitranscriptome modifications in virus replication.

The authors concluded that the observed differences between the viruses are consistent with HCMV genes aligning with differentiation codon usage and SARS-CoV-2 genes reflecting proliferation codon usage. This observation's connection to the biology of HCMV and SARS-CoV-2 lies in the codon usage of structural and gene expression-related viral genes, showing a significant adaptation to host cell tRNA pools. Notably, these genes from both viruses demonstrated the highest adaptation to the tRNA pool of infected cells. The reason behind this phenomenon remains unclear. One hypothesis suggests that a high level of structural gene expression is necessary during activation. Testing this hypothesis could involve examining if hindering tRNA modifications affects virus morphogenesis. In summary, this study presents an interesting and innovative perspective on how viruses modify the translation machinery. The meticulous analysis sheds light on a central interaction point between viruses and their host cells.

2. Significance:

Significance (Required)

In summary, this study presents an interesting and innovative perspective on how viruses modify the translation machinery. The meticulous analysis sheds light on a central interaction point between viruses and their host cells.

3. How much time do you estimate the authors will need to complete the suggested revisions:

Estimated time to Complete Revisions (Required)

(Decision Recommendation)

Between 1 and 3 months

Yes

Review #3

1. Evidence, reproducibility and clarity:

Evidence, reproducibility and clarity (Required)

****Summary****

Aharon-Hefetz et al. present the expression dynamics and modification signatures of tRNAs using DM-tRNA-seq in human foreskin fibroblasts or Calu3 cells during infections with two diverse viruses, HCMV and SARS-CoV2, respectively. They also use a newly designed tRNA-centric CRISPR library to screen the essentiality of tRNA and tRNA factors during HCMV-GFP infection. They find several tRNAs that are differentially expressed during HCMV infection, and most closely resemble the set of tRNAs shown to be used during cellular differentiation. Additionally, tRNA differential expression does not resemble that following interferon treatment, implying that virus modulation of tRNAs is unique to the general interferon response. They compare codon usage signatures during infection to their prior-defined sets of proliferation/differentiation tRNA genes. In their CRISPR screen, they

find that different tRNAs can promote or restrict HCMV infection levels, as measured by the intensity of GFP fluorescence marker in their virus. Surprisingly, there were few tRNA modification factor hits that contributed to growth or infection.

****Major Comments****

1. The topic of this work is important, and the analysis performed here is assumed to be top quality, based on the previous work by the last author. The weakness with this body of work is a lack of rigor, specifically regarding validation and follow-up studies. Without these experiments, the reader lacks confidence in stated conclusions. For example:

- a. There is no validation or clue to how penetrant CRISPR is against tRNA genes. Given how duplicated some tRNA families are, it is possible that CRISPR is more effective against certain families compared to others. While this is likely an inherent caveat in all CRISPR screens, it would lend confidence in this approach to see some validation of tRNA KO by northern blot or RT-qPCR or sequencing.

- b. There is no validation that tRNA modification factor knockouts alter tRNA modification levels. Without this knowledge, the lack of essentiality cannot be confidently and fully interpreted. If the group does not validate whether individual tRNA modification factor knockouts alter modification profiles, then all possible explanations should be posited. For example, it is possible that 1) there could be major redundancy among tRNA modification enzymes, as the authors posit in the discussion 2) tRNA modification enzymes are not essential for growth bc their activity/the modification they place is non-essential for growth, OR 3) the knockouts are not fully penetrant. I think this discussion should be expanded to make caveats clearer. Perhaps referencing whether tRNA modification factors have been shown to be essential in other CRISPR screens would be helpful.

- c. There is no validation that factors modulating GFP intensity in the HCMV screen actually impact virus replication. This is the point most important to this body of work. While GFP intensity does correlate to genome copies as shown by the authors, GFP read-out on a case-by-case basis could be simply due to factors required for expression/translation of GFP. Are any of the tRNA hits enriched or not represented in GFP reporter sequence? Either way, this information is informative. Additionally, given that the hits are cross-compared ONLY to other infected (low intensity "GFP+") cells, and not to an uninfected population, there is no guarantee that these primarily drive HCMV infection. The top hits should be validated in HFFs, infected with HCMV, with resulting titers/viral gene expression/genome copies measured. Additionally, the reasons for not using a GFP-population as a control should be clarified.

2. Though careful codon usage analysis for HCMV versus the human host was analyzed, it seems pertinent to analyze whether the differentially expressed tRNAs during infection

correlate to either codon usage profiles. Figure 3C and S3C intend to address this point for viral gene groups; however, I would encourage the authors to expand the description of these results to make them easier to interpret, especially for those not in the tRNA field. For example, "tRNA adaptation index (tAI)" is not defined in the text, but simply referenced. For clarity, you should include a brief explanation of what this measure describes. Following, when reporting results from Figure 3, the results can then be delivered with more specific and interpretable language. These steps will ensure maximal scientific communication to the audience. Finally, given that changes are most visible at 72 hpi, the analysis should include expression based on this time point for comparison.

****Minor Comments****

1. I would recommend more care in terminology used for the CRISPR screen (Figures 5 and 6) to make the manuscript easier to digest. Labeling sgRNAs-containing cells as "Reduced Growth/Infection" or "Increased Growth/Infection" is not immediately easy to understand. For example, saying this sgRNA "increased growth" could refer to the knockdown increasing growth OR could mean that this sgRNA was enriched in cells with increased growth, which are opposing. It might be more clear to state to use depleted/enriched terminology in these figure labels. This also applies to the text, be sure to plainly describe the terminology and what it means each time you refer to the CRISPR results.
2. Is there actual evidence that the new tRNA sgRNA library is more effective than that used previously? State if so.
3. Fig 1A-C: The cutoff for "red" symbol distinction is not stringent enough. 1.05 would be red, but that is not convincingly upregulated. The cutoff should be at least $FC > 1.2$.
4. Need thorough description of tRNA bioinformatics and modification analysis (citing past work is not appropriate here-need to make accessible to your audience)
5. Line 182- Result headings could be more informative, even with small adjustments. For example "Specific tRNA modifications are modulated in response to HCMV infection" is more clear and accurate, as there are only a few measurable changes in tRNA modification. Limitations of using sequencing techniques to analyze modifications (versus MS) should also be discussed.
6. It is not immediately clear why the viral plot looks different in Fig S3B compared to Fig 3B.
7. Line 254-255. This point is not immediately clear-please include more specific language detailing the logic leading to this conclusion.
8. Line 408- "may be essential"-I would modify the language here. Especially given there is no true comparison with uninfected cells.
9. There are a number of recent publications profiling tRNA expression in herpesviruses. These should be mentioned and discussed in the context of this work. I know some were

included in the reference list, but the body of work as a whole, and how this work fits in and pushes the horizon, could be further emphasized. It is quite impressive that this is a conserved feature of herpesvirus infection.

- a. PMID: 36752632

- b. PMID: 35110532

- c. PMID: 34535641

- d. PMID: 33986151

- e. PMID: 33323507

- f. PMID: 35458509

10. CoV2 discussion point-The lack of tRNA expression regulation might have more to do with the length of the infection (6 hpi cov2- also didn't see much a change at 5hpi with hcmv). This should be proposed as a possibility.

11. Line 582. Misspelled schlafen in discussion. (SLFN, not SFLN)

2. Significance:

Significance (Required)

General assessment: I found this paper exciting to read, given the dearth of knowledge regarding viral modulation of tRNA expression. However, the work is highly descriptive, with a complete absence of follow-up or validation studies. At the very least, I would have hoped that the authors validated that viral titer (and not just GFP intensity) was impacted by some of the hits. The lack of confirmation and quality control overall diminishes confidence in the stated conclusions. However, I think the topic is timely, important, and that this manuscript offers tools to the community at large to learn more about viral manipulation or other drivers of tRNA regulation. Once follow-up/validation experiments are added to the work, as detailed below, this manuscript will be of broad importance and highly impactful.

Advance: While there have been many studies suggesting tRNA regulation occurs during viral infection (these pubs should be referenced as mentioned above), this is an advance due to the fact that it begins to address whether tRNA expression changes functionally impact viral replication. This will be much more solid with follow-up experiments confirming that hits alter HCMV replication (rather than GFP intensity).

Audience: This will be of broad interest to those with interest in virology and gene expression. The new sub-libraries of tRNA-related factors might be useful to be tested in other cell types and settings. Again, as the work stands, it is descriptive and hypothesis-stimulating, but the conclusions need validation and further support.

3. How much time do you estimate the authors will need to complete the suggested revisions:

Estimated time to Complete Revisions (Required)

(Decision Recommendation)

More than 6 months

Yes

Revision Plan

Manuscript number: RC-2024-02487R

Corresponding author(s): Noa Aharon-Hefetz, Michal Schwartz, Orna Dahan, Noam Stern-Ginossar, Yitzhak Pilpel

1. General Statements

Dear Madan,

We have recently submitted our paper titled "Essentiality and dynamic expression of the human tRNA pool during viral infection" to Review Commons. The paper was reviewed rather positively, although some useful comments, concerns, and advice came up. We have started a revision process aimed at addressing all points. We already addressed the majority of the points – all those that did not involve a new experiment. In parallel, we are preparing for an experiment that was suggested by Referee #3 that could substantiate and provide further gene-specific support to the CRISPR library-based screen. While we believe that the library screen we performed is valid and adheres to the convention in the field, we are happy to update with individual gene tests when we have them.

In any case, we believe that Mol. Sys Biol. would ultimately be the most suitable journal for our paper. While the work addresses viral biology, we take a systems-level approach of the style often published in your journal.

Thus, after consultation with RC's editor, I'm writing to you with 1) a detailed point-by-point with a plan for a new suggested experiment and 2) a revised version of the paper accordingly, in which we have already addressed most of the points (but the additional suggested experiment)

We would appreciate your opinion at this stage so we can make a plan.

With best regards,

Tzachi.

2. Description of the planned revisions

Reviewer #3- major comments

1. The topic of this work is important, and the analysis performed here is assumed to be top quality, based on the previous work by the last author. The weakness with this body of work is a lack of rigor, specifically regarding validation and follow-up studies. Without these experiments, the reader lacks confidence in stated conclusions. For example:
 - a. There is no validation or clue to how penetrant CRISPR is against tRNA genes. Given how duplicated some tRNA families are, it is possible that CRISPR is more effective against certain families compared to others. While this is likely an inherent caveat in all CRISPR screens, it would lend confidence in this approach to see some validation of tRNA KO by northern blot or RT-qPCR or sequencing.

We thank the reviewer for raising this important issue. Indeed, many tRNA genes appear in multiple copies in the human genome. Yet, based on our previous work, we expect parallel editing of

Revision Plan

multiple copies using the same sgRNA. In our previous work (doi.org/10.7554/eLife.58461), we validated, based on several tRNA families, the ability of our tRNA CRISPR system to successfully target and affect tRNA expression levels. This included sequencing of the edited tRNA genes (i.e., DNA sequencing), in which we observed diverse INDEL mutations that predicted full disruption of the tRNA structure. Furthermore, we sequenced the tRNA pool of CRISPR-edited cells and found the downregulation of the targeted tRNAs to be up to 2-4-fold. This previous work provides foundations and confidence in this tRNA-CRISPR approach.

Nevertheless, to further mitigate the reviewer's concern, we also plan to perform additional experiments in the current settings. We will choose individual tRNAs from our CRISPR screen as representatives to validate CRISPR editing. We will target each tRNA independently and test expression reduction by sequencing. We shall share the results in the full revision if granted.

Additionally, given that the hits are cross-compared ONLY to other infected (low intensity "GFP+") cells, and not to an uninfected population, there is no guarantee that these primarily drive HCMV infection. The top hits should be validated in HFFs, infected with HCMV, with resulting titers/viral gene expression/genome copies measured.

We agree that additional experiments on some hits may be warranted. We plan to examine for such an effect on infection using an individual gene version of the assay. In particular, we will target individually candidate tRNA genes following validation (as described in the point above). We will then infect the tRNA-targeted cells with HCMV and measure the effectiveness of HCMV infection using a standard titer assay.

3. Description of the revisions that have already been incorporated in the transferred manuscript

Reviewer #1 (Evidence, reproducibility and clarity (Required)):

I have mixed feelings regarding this manuscript. On the one hand, the authors did an impressive amount of work. On the other hand, the manuscript seems overly descriptive (writing should be more concise) without a clear message or hypothesis that is cohesive to all the presented evidence. Below, I will outline my concerns.

We appreciate the comment about missing a cohesive presentation. We worked extensively to improve that in the revised manuscript.

Reviewer #1- first part

1. I am not an expert in the field of viral biology and immunology. I wonder how well the IFN treatment mimics the cellular response to infection (yet without the virus). Also, how good is ruxolitinib at blocking the IFN response? I would appreciate it if you could explain both with one or two sentences and provide the necessary references.

The reviewer is correct that we cannot claim that interferon treatment mimics exactly the cellular response. However, the expression of interferon-stimulated genes (ISGs) is a major arm of the

antiviral response to HCMV (c.f. [doi:10.3390/v10090447](https://doi.org/10.3390/v10090447), [doi:10.2217/fvl-2018-0189](https://doi.org/10.2217/fvl-2018-0189)). In addition, Ruxolitinib is a potent and selective Janus kinase 1 and 2 inhibitor ([doi:10.1021/ol900350k](https://doi.org/10.1021/ol900350k)), and we have shown in the past that it very effectively reduces the expression of many ISGs ([doi:10.1038/s41590-018-0275-z](https://doi.org/10.1038/s41590-018-0275-z)). Since ISGs constitute a major part of the host response to HCMV infection, the fact that their expression leads to minor changes in the tRNA pool strongly suggests that it is mainly the virus (as opposed to the host cell) that mediates the changes seen in the tRNA pools during HCMV infection. In the revised version, these claims were amended, and relevant references were added (pages 5, lines 132-136).

2. (MAJOR) Can these two treatments really allow the effects of host response and viral infection to be separated? OR in other words, are these two effects really orthogonal? In my opinion, they are NOT. Fig. 1E seems to support my opinion, as the changes seen for the "IFN" sample relative to the "uninfected" sample (referred to as "changes-A" below), are parallel to the changes seen for the "24hpi + ruxo" sample relative to the "24hpi" sample ("changes-B"). More specifically, changes-A represent the host response, as argued by the author, whereas changes-B represent the elimination of the host response (due to ruxo, conditioned on the virus-driven effect). If the virus-driven effect and the host response could really be separated, one would expect changes-A and changes-B are more or less opposite. However, they appeared to be parallel, suggesting that uninfected versus infected conditions can have totally different (even opposite) host responses. More importantly, if one cannot separate the host response from virus-driven effects, the conclusion of "tRNA changes are driven by virus, not host response" is then unfounded.

This is an important point to clarify. Changes-A indeed represent the effect of the host antiviral response on the tRNA pool. Changes-B, however, represent a mix of two effects. 1: counteracting the effect of the host antiviral response on the tRNA pool, which we show is a minor effect, and 2: The enhanced effect of the virus, since ruxolitinib, by inhibiting the host antiviral response, enhances the viral infection. It may indeed be that both the virus and the host antiviral effects are in the same direction. However, it is clear that the antiviral effect is minor. Thus, it is likely that the second effect of ruxolitinib (i.e., allowing enhanced viral infection) is the more substantial one. Therefore, it seems as though the viral effect and the elimination of the host effect are in the same direction. This point was clarified in the revised version (page 6, lines 145-146).

3. Even if we let go of this previous point and accept that these results indeed offer some support for the notion that the virus-driven effect are the main contributor to the shifts in tRNA pool, the support is at best moderate. A big gap here is "how?" I suggest the authors should at least give some insight on how virus can do that in Discussion (and mention it with one sentence in Results).

We certainly welcome the challenge, which we now meet in the revision. In short, here, transcription regulation of tRNAs, mainly upon viral infection, is poorly studied. Unlike other herpesviruses, HCMV does not cause a host shut-off of the host transcripts. Upon HCMV infection, the tRNA transcription machinery is upregulated significantly, which probably contributes to the

Revision Plan

upregulation in pre-tRNA (doi.org/10.1016/j.semcdb.2023.01.011). However, it is still unknown what the viral factors are that promote upregulation in the tRNA transcription machinery. We now relate to this point in the results (page 6, lines 147-148) and discuss the known effects of viral infection of tRNA expression in the discussion section (page 15, lines 447-451).

4. The authors compared the HCMV codon usage to the proliferation and differentiation signatures of human cells. But these two signatures are not compared with measured tRNA expression. It might shed some light on the general characteristics of tRNA pool shifts due to infection (towards a proliferation-like or differentiation-like signature). This fits in the general topic of virus-host interaction and might give more evidence for the point that HCMV is adapted to a differentiation signature (as it drives the host into that state).

We performed the analysis suggested by the reviewer. We found that the tRNA pool of uninfected HFF cells correlated to the same extent with proliferation codon usage ($r=0.29$, $p\text{-value}=0.029$) and differentiation codon usage ($r=0.26$, $p\text{-value}=0.05$). Similar correlations to the proliferation and differentiation signature were found when analyzing the tRNA pool 72h post-infection (proliferation $r=0.33$, $p\text{-value}=0.011$, differentiation $r=0.28$, $p\text{-value}=0.034$). This result suggests no general shift in the tRNA pool towards a specific codon usage signature.

5. How is the dashed box in Fig3A/B chosen?

We determined the dashed lines based on the most prominent groups of transcripts best adapted to proliferation or differentiation codon usage signatures. Figure S3A clearly shows the two groups without viral genes. We emphasize this point in the legend of Figure S3A (page 36, lines 1157).

6. The tAI values shown in Fig3C-E are extremely low (compared to other reports I am aware of). Does this mean that the adaptation of viral codon usage to human cell supply is actually very weak? This is in opposition to the major claims made in this section.

We acknowledge that the tAI values presented here are lower than typically presented. However, this is due to how tAI was calculated rather than the potential weak adaptation between viral genes and tRNA supply. Specifically, unlike previous works that estimate tRNA availability based on tRNA gene copy number, here we calculated tAI using tRNA sequencing (in order to capture the dynamics in the tRNA pool during infection). Indeed, the value of tAI calculated by tRNA read counts is lower than tAI calculated by tRNA copy number. This is due to the skewed distribution of tRNA read counts (some tRNAs are highly expressed, and others are lowly expressed), while tRNA copy number is distributed more evenly. Thus, due to the mathematical nature of the tAI (computing geometric rather than arithmetic average of tRNA availability), the skewed distribution observed in the data results in lower tAI values. When computing tAI based on gene copy number, we get higher tAI values (0.3 on average). Nevertheless, as all tAI calculations here were done similarly, the comparisons between gene groups or genes are valid.

7. I believe that the part about SARS-CoV-2 could be made more concise. It is sufficient to mention that results may differ from those obtained with HCMV in one paragraph.

Revision Plan

The section on SARS-Cov-2 is now made rather succinct. This virus is mainly given as a comparison to the primary virus studied in this paper - HCMV.

8. Line 299 on page 11 - I do not believe codon usage between different viruses can be directly compared, let alone reaching such a conclusion. Some viruses have low CAI or tAI to humans, but they have co-evolved with humans for a long time. Furthermore, there are viruses that infect multiple hosts, but their CAI for a host with which they have long co-evolved is higher while their CAI for a host that is relatively new is lower.

We agree with the reviewer that a direct link between co-evolution time and tAI may not always exist. Indeed, other factors might explain the observation that SARS-CoV-2 genes are less adapted than HCMV genes. These may include effective population sizes and mutation rates that vary substantially. We, therefore, removed this conclusion from the manuscript.

9. (MAJOR) A more general comment is that there is a difference between tRNA expression and the abundance of translation-ready tRNA. The process of charging tRNA with amino acids may take a long time. It is the abundance of the charged-tRNA (the ternary complex of aminoacylated tRNA and EF-Tu-GTP) that is of biological importance. In this regard, the use of tRNA expression falls short.

The reviewer raises a valid point. Indeed, our tRNA sequencing protocol measures both charged and uncharged tRNAs that constitute the cell's mature tRNA pool. Compared to previous studies that focus on the transcription process of tRNAs in viral-infection models by sequencing the pre-tRNAs, here we look at the mature tRNA pool that accounts for both transcription and post-transcription processes. Therefore, we changed the use of "tRNA expression" to "mature-tRNA levels" and "highly" or "lowly-abundant tRNAs" rather than "highly" or "lowly expressed tRNAs" in the manuscript. We note, however, that although limited in the ability to differentiate between charged and uncharged tRNAs, the tRNA sequencing protocol used here is commonly used and validated as a state-of-the-art protocol in tRNA sequencing ([10.1016/j.molcel.2021.01.028](https://doi.org/10.1016/j.molcel.2021.01.028), [10.1038/s41467-020-17879-x](https://doi.org/10.1038/s41467-020-17879-x), etc.), mainly because it addresses the level of "ready-to-use" tRNA.

Reviewer #1- second part

1. (MAJOR) Prior to the actual competition assay in the first high-throughput screen (cell competition assay), the authors applied two days of antibiotic selection and two days of recovery. This could result in a serious problem of false negatives or drop outs. Specifically, an sgRNA targeting an essential gene with high efficiency would kill the cells, leaving no (or a small number of) cells in the ancestor population at the beginning of the competition process. A sgRNA's enrichment in competing populations cannot be reliably estimated in such situations. I am not certain that the FDR used in Figure 5B is sufficient to address this issue. Please clarify whether it could. Providing raw counts for competing and ancestor populations would also be helpful.

As customary in CRISPR screens, the step of lentiviral transduction and antibiotic selection is necessary to ensure that only CRISPR-edited cells are left in the population. Indeed, essential genes

Revision Plan

like housekeeping genes are probably removed from the competing population relatively quickly, which might result in their dropouts. We could have lost some tRNA hits in the cell growth CRISPR screen (Figure 5B-C) because of their overall essentiality for cell growth. The MAGeCK tool we used, the state-of-the-art in the field, filters out sgRNAs with low read counts to be able to calculate false discovery rates. Indeed, we identified 15 tRNAs that were depleted from the competing cells. We believe that our procedure minimizes the concern of dropouts. tRNA dropout in the HCMV infection CRISPR screen (Figure 6B-C) can also happen, which means our screen underestimates the essentiality of tRNAs to HCMV infection. However, this concern does not affect the significance of the hits we did find. We acknowledge this inherent difficulty in CRISPR screens and will provide the raw read counts of all samples upon full submission. We emphasize, though, that while valid, this concern applies to essentially any CRISPR screen that is commonplace in genomics these days.

2. It is also highly questionable to me the nearly negligible effects of tRNA modification enzymes. This may be explained by the point above. Indeed, the dots of tRNA modification enzymes in general appear to have higher FDR (lower y values) when compared to red dots with similar enrichment levels.

This is a valid point. We found a lack of essentiality of tRNA modification enzymes in both screens. We analyzed additional CRISPR screens and compared the effect of tRNA modification enzyme knockouts relative to the restriction and dependency factors we used in the library. The tested screens included 34 knockout CRISPR screens we downloaded from the BioGRID ORCS database that have similar parameters to our screen. Namely, they all test cell proliferation in a time-course manner, using a pooled sgRNA library and using the MAGeCK tool for data analysis. Overall, the screens use different human cell lines and diverse sgRNA libraries. Although potentially surprising, we found that the lack of essentiality of tRNA modification enzymes was also observed in the analyzed CRISPR screens (Figure S5B and on page 11, lines 322-330, and on page 18, lines 539-541). One potential reason is if some of these enzymes were "backed up" by others, which we mentioned. Another explanation is that most tRNA modification enzymes are indeed not essential for growth and for viral infection (now described in the Discussion, page 18, lines 544-545). Alternatively, dropouts can explain this result, as suggested by the reviewer. To examine the likelihood of the dropout option, we examined the average raw read count of the tRNA modification enzyme in the ancestor samples. We compared it to that of other sub-groups. We found that raw read counts of the tRNA modification enzymes are not different than other sub-groups in the CRISPR library. Thus, the dropout issue cannot explain our screens' lack of essentiality of tRNA modification enzymes.

3. The screen based on IE2-GFP labeled HCMV measures a phenotype that is very difficult to interpret. Particularly, I am not sure if GFP2 and GFP3 are good controls for comparing GFP4 (GFP1 might be better). Various factors can affect GFP levels, including, but not limited to, dilution caused by a rapidly dividing host cell, unhealthy translational machinery resulting from infection or microenvironment. My point is supported by some observations in Fig6B. For example, SEC61B, a restriction factor for HCMV infection, is enriched in the GFP2 group,

contrary to expectations. It is necessary for the authors to prove with firm evidence that their choice of GFP signal thresholds is appropriate.

We acknowledge the concern. Specifically, the translation of the GFP gene itself could be affected by the tRNA manipulation done. To account for this potential concern, we tested the codon usage of the eGFP gene (which is the GFP version we used in the system) and compared it with tRNA essentiality, as determined by the cell growth CRISPR screen. We report this in the revised manuscript (page 13, lines 390-392, and added Figure S6D). We found that GFP does not tend to significantly use codons that correspond to essential or less essential tRNAs. The same lack of correlation was also found for the tRNA essentiality upon HCMV infection (not shown).

More generally, we show that GFP intensity does correlate with viral genome copies (Figure S6A). Also, from mRNA-seq data of temporal HCMV infection ([10.1016/j.celrep.2022.110653](https://doi.org/10.1016/j.celrep.2022.110653)), IE2 (UL122) shows a dynamic expression- high expression peak in early infection, then a decline in expression level followed by a gradual increase.

Dynamic expression of UL122 (IE2) during HCMV infection. The X-axis depicts the time point along HCMV infection, and the y-axis depicts the normalized mRNA expression of UL122.

Altogether, we believe that the IE2-GFP level provides a good estimation for viral load.

Regarding SEC61B, which served as a control in our screen – the referee is rightly asking why it behaves oppositely from what's expected, given that this was supposed to be a restriction factor of HCMV infection. We returned to the literature on the essentiality of this gene upon HCMV infection. In Weissman's paper ([10.1038/384432a0](https://doi.org/10.1038/384432a0)), which was the reference for choosing control genes in our system, this gene was targeted through two different CRISPR technologies, once with CRISPR knockout and once with CRISPRi. Interestingly, only upon CRISPRi did this gene prove to be a restriction factor (i.e., improved infection upon reduction of the gene). We comment on this peculiar fact in the revised manuscript (page 13, lines 370-374). However, we note that the rest of our positive and negative controls deliver the expected results – increasing or reducing infection as expected from their role, thus lending considerable support to our experimental system. It is possible, especially in light of our screen, and since other positive and negative controls behave as expected, that the status of the SEC61B gene as a "restriction factor" of viral infection needs to be reconsidered, as we now suggest.

4. I would appreciate more information regarding why restriction factors of cell growth have a high GFP2/GFP4. Intuitively, a KO of restriction factors of cell growth should result in better growth and higher GFP, thus leading to enrichment in GFP4, not GFP2.

Revision Plan

The reviewer raises an interesting question (although not at the heart of this work, as sgRNAs for the cell growth restriction factor mainly aim to serve as controls for the CRISPR screen). HCMV has a complex interaction with the cellular cell cycle. Specifically, it establishes a unique G1/S arrest that is both stimulatory and inhibitory since, on the one hand, it serves the virus to arrest the cell cycle, a critical step for viral genome replication. On the other hand, the virus needs many of the resources that serve cell growth. Both p53 and CDKN1A are important regulators at this stage; therefore, their interaction with the virus may indeed be complex. For example, p53 is upregulated by a viral infection. However, it is sequestered in the viral replication compartments, and its transcriptional are down-regulated, but its absence harms viral propagation ([doi: 10.1128/mBio.02934-21](https://doi.org/10.1128/mBio.02934-21), [doi: 10.1128/jvi.72.3.2033-2039.1998](https://doi.org/10.1128/jvi.72.3.2033-2039.1998), [doi: 10.1128/jvi.00505-06](https://doi.org/10.1128/jvi.00505-06)). Therefore, it is not surprising that genes related to cell growth and cell cycle have complex effects on HCMV infection. We mention the essentiality of p53 for HCMV infection in the results (page 14, line 404).

5. Line 404 "nonetheless"

We appreciate the reviewer for noticing the typo. We corrected it.

Reviewer #1 (Significance (Required)):

The relation between human tRNA supply and viral translation is a topic of profound biological and biomedical importance. In this study, the authors used HCMV infection as the primary model to investigate this question. Results fall into two major parts: (i) changes in the tRNA pool during viral infection, and (ii) the impact of tRNA-related gene KO on viral infection.

We appreciate the detailed report. We addressed the major points raised in the revised manuscript.

Reviewer #2 (Evidence, reproducibility and clarity (Required)):

In this study by Aharon-Hefetz et al., the researchers examined changes in tRNA pools during virus infections. The translation machinery plays a crucial role in virus replication. Consequently, host cells have developed sensors and effectors within this compartment to counteract viral mechanisms. The translation apparatus serves as a pivotal point in the virus-host conflict. Therefore, investigating alterations in the translation machinery during infections is vital for gaining a comprehensive understanding of the infection process.

This study offers a thorough and high-quality analysis of data in a relevant cell culture system involving two different viruses. By conducting tRNA sequencing, the researchers studied the human tRNA pool following infections with human Cytomegalovirus (HCMV) and SARS-CoV-2. Changes in tRNA expression induced by HCMV were mainly driven by the virus infection itself, with minimal impact from the cellular immune response. Interestingly, specific tRNA post-transcriptional modifications seemed to influence stability and were subject to manipulation by HCMV. Conversely, SARS-CoV-2 did not lead to significant alterations in tRNA expression or post-transcriptional modifications.

Moreover, a systematic CRISPR screen targeting human tRNA genes and modification enzymes allowed the identification of specific tRNAs and enzymes that either enhanced or reduced HCMV infectivity and cellular growth. This information enabled them to control the development of HCMV-specific tRNA modifications, highlighting the importance of these tRNA epitranscriptome modifications in virus

Revision Plan

replication.

The authors concluded that the observed differences between the viruses are consistent with HCMV genes aligning with differentiation codon usage and SARS-CoV-2 genes reflecting proliferation codon usage. This observation's connection to the biology of HCMV and SARS-CoV-2 lies in the codon usage of structural and gene expression-related viral genes, showing a significant adaptation to host cell tRNA pools. Notably, these genes from both viruses demonstrated the highest adaptation to the tRNA pool of infected cells. The reason behind this phenomenon remains unclear. One hypothesis suggests that a high level of structural gene expression is necessary during activation. Testing this hypothesis could involve examining if hindering tRNA modifications affects virus morphogenesis.

In summary, this study presents an interesting and innovative perspective on how viruses modify the translation machinery. The meticulous analysis sheds light on a central interaction point between viruses and their host cells.

Reviewer #2 (Significance (Required)):

In summary, this study presents an interesting and innovative perspective on how viruses modify the translation machinery. The meticulous analysis sheds light on a central interaction point between viruses and their host cells.

We thank the reviewer for finding our work interesting, innovative, and well analyzed

Reviewer #3 (Evidence, reproducibility and clarity (Required)):

Summary

Aharon-Hefetz et al. present the expression dynamics and modification signatures of tRNAs using DM-tRNA-seq in human foreskin fibroblasts or Calu3 cells during infections with two diverse viruses, HCMV and SARS-CoV2, respectively. They also use a newly designed tRNA-centric CRISPR library to screen the essentiality of tRNA and tRNA factors during HCMV-GFP infection. They find several tRNAs that are differentially expressed during HCMV infection, and most closely resemble the set of tRNAs shown to be used during cellular differentiation. Additionally, tRNA differential expression does not resemble that following interferon treatment, implying that virus modulation of tRNAs is unique to the general interferon response. They compare codon usage signatures during infection to their prior-defined sets of proliferation/differentiation tRNA genes. In their CRISPR screen, they find that different tRNAs can promote or restrict HCMV infection levels, as measured by the intensity of GFP fluorescence marker in their virus. Surprisingly, there were few tRNA modification factor hits that contributed to growth or infection.

Reviewer #3- major comments

1. b. There is no validation that tRNA modification factor knockouts alter tRNA modification levels. Without this knowledge, the lack of essentiality cannot be confidently and fully interpreted. If the group does not validate whether individual tRNA modification factor knockouts alter modification profiles, then all possible explanations should be posited. For example, it is possible that 1) there could be major redundancy among tRNA modification enzymes, as the authors posit in the Discussion 2) tRNA modification enzymes are not essential for growth bc their

Revision Plan

activity/the modification they place is non-essential for growth, OR 3) the knockouts are not fully penetrant. I think this Discussion should be expanded to make caveats clearer. Perhaps referencing whether tRNA modification factors have been shown to be essential in other CRISPR screens would be helpful.

Regarding the possible explanations for the lack of essentiality of tRNA modification enzymes, we agree with all three possibilities the reviewer raised. Reviewer #1 raised an additional option, in which tRNA modification enzymes are essential for HCMV infection and cell growth; thus, we cannot detect them in the screens because they drop out early in the process (before collecting the ancestor samples). We checked this possibility and found comparable read counts of sgRNAs targeting tRNA modification enzymes to that of other sub-libraries. This result suggests the drop-outs of sgRNA targeting are unlikely to happen on our screens. Furthermore, as the reviewer asked, we analyzed additional CRISPR screens and compared the effect of tRNA modification enzyme knockouts relative to the restriction and dependency factors we used in the library. The tested screens included 34 knockout CRISPR screens we downloaded from the BioGRID ORCS database that have similar parameters to our screen. Namely, they all test cell proliferation in a time-course manner, using a pooled sgRNA library and using the MAGECK tool for data analysis. Overall, the screens use different human cell lines and diverse sgRNA libraries. Although potentially surprising, we found that the lack of essentiality of tRNA modification enzymes was also observed in the analyzed CRISPR screens (Figure S5B and on page 11, lines 322-330, and on page 18, lines 539-541).

c. There is no validation that factors modulating GFP intensity in the HCMV screen actually impact virus replication. This is the point most important to this body of work. While GFP intensity does correlate to genome copies as shown by the authors, GFP read-out on a case-by-case basis could be simply due to factors required for expression/translation of GFP. Are any of the tRNA hits enriched or not represented in GFP reporter sequence? Either way, this information is informative.

We acknowledge the concern. Specifically, the translation of the GFP gene itself could be affected by the tRNA manipulation done. To account for this potential concern, we tested the codon usage of the eGFP gene (which is the GFP version we used in the system) and compared it with tRNA essentiality, as determined by the cell growth CRISPR screen. We report this in the revised manuscript (page 13, lines 390-392, and added Figure S6D). We found that GFP does not tend to significantly use codons that correspond to essential or less essential tRNAs. The same lack of correlation was also found for the tRNA essentiality upon HCMV infection (not shown).

Additionally, the reasons for not using a GFP- population as a control should be clarified.

The reviewer suggests comparing GFP1/2/3 to an ancestor in addition to comparing them to GFP4. Towards that we now show a GFP2 vs ancestor comparison (shown below). The results look very similar and are now added to the supplemental material of the revised manuscript (page 13, lines 385-387, Figure S6B).

2. Though careful codon usage analysis for HCMV versus the human host was analyzed, it seems pertinent to analyze whether the differentially expressed tRNAs during infection correlate to either codon usage profiles. Figure 3C and S3C intend to address this point for viral gene groups; however, I would encourage the authors to expand the description of these results to make them easier to interpret, especially for those not in the tRNA field. For example, "tRNA adaptation index (tAI)" is not defined in the text, but simply referenced. For clarity, you should include a brief explanation of what this measure describes. Following, when reporting results from Figure 3, the results can then be delivered with more specific and interpretable language. These steps will ensure maximal scientific communication to the audience.

We appreciate the reviewer's comment regarding the importance of scientific communication and making this manuscript easier to interpret, especially for readers unfamiliar with the world of tRNAs and translation efficiency. We added a description of our motivation to use tAI and the meaning of the measurement (page 9, lines 241-243). We also elaborated on the results part and made the results more interpretable (page 9, lines 245 and 249-250).

Finally, given that changes are most visible at 72 hpi, the analysis should include expression based on this time point for comparison.

Regarding the time point used for tAI calculation (Figure 3), we tested the tAI measured by the tRNA pool at 72hpi and got very similar results to that obtained using the tRNA pool measured at 24hpi. As 24hpi represents the pick of HCMV infection, we decided to present this analysis. In the current revised version, we also added the analysis done using the tRNA pool measured 72hpi as suggested by the reviewer (Figure S3D).

Reviewer #3- minor comments

1. I would recommend more care in terminology used for the CRISPR screen (Figures 5 and 6) to make the manuscript easier to digest. Labeling sgRNAs-containing cells as "Reduced Growth/Infection" or "Increased Growth/Infection" is not immediately easy to understand. For example, saying this sgRNA "increased growth" could refer to the knockdown increasing growth OR could mean that this sgRNA was enriched in cells with increased growth, which are opposing. It might be more clear to state to use depleted/enriched terminology in these figure labels. This also applies to the text, be sure to plainly describe the terminology and what it means each time you refer to the CRISPR results.

This is a good point. Indeed, focusing on the significant enrichment of the sgRNAs, rather than their effect on growth or infection, is more straightforward. We changed the terminology in Figures 5C and 6C and the text in the current version.

2. Is there actual evidence that the new tRNA sgRNA library is more effective than that used previously? State if so.

We assume the referee refers to our previous paper on the smaller-scale library (doi.org/10.7554/eLife.58461). The addition here is that the library is much more comprehensive

Revision Plan

(the previous one targeted only 20 tRNAs). We point it out in the revised manuscript (page 17, lines 499-501).

3. Fig 1A-C: The cutoff for "red" symbol distinction is not stringent enough. 1.05 would be red, but that is not convincingly upregulated. The cutoff should be at least $FC > 1.2$.

We thank the reviewer for bringing our attention to this point. In the current version, we changed the cutoff of absolute fold change higher than 1.2 in Figures 1A-C and S1A (also in legend).

4. Need thorough description of tRNA bioinformatics and modification analysis (citing past work is not appropriate here-need to make accessible to your audience).

Further thorough descriptions of tRNA bioinformatics and modification analysis are added in the revised version (page 6, lines 149-151, page 7, lines 178-183).

5. Line 182- Result headings could be more informative, even with small adjustments. For example "Specific tRNA modifications are modulated in response to HCMV infection" is more clear and accurate, as there are only a few measurable changes in tRNA modification. Limitations of using sequencing techniques to analyze modifications (versus MS) should also be discussed.

We changed that heading accordingly.

We also mentioned the advantages and disadvantages of using sequencing to assess tRNA modification levels (page 7, lines 184-187).

6. It is not immediately clear why the viral plot looks different in Fig S3B compared to Fig 3B.

We thank the referee for spotting this. We employed different length cutoffs on the genes in each panel and have now fixed that in the revised manuscript.

7. Line 254-255. This point is not immediately clear-please include more specific language detailing the logic leading to this conclusion.

Indeed, the logic here was missing. The idea was that longer genes are associated with gene conservation, hence functionality. Thus, non-canonical HCMV genes that are both long and codon-optimized might have a function during HCMV infection. We added this explanation to the text (pages 8-9, lines 235-238).

8. Line 408- "may be essential"-I would modify the language here. Especially given there is no true comparison with uninfected cells.

We improved the language throughout the revised manuscript.

9. There are a number of recent publications profiling tRNA expression in herpesviruses. These should be mentioned and discussed in the context of this work. I know some were included in the reference list, but the body of work as a whole, and how this work fits in and pushes the horizon, could be further emphasized. It is quite impressive that this is a conserved feature of herpesvirus infection.

- a. PMID: 36752632

Revision Plan

b. PMID: 35110532

c. PMID: 34535641

d. PMID: 33986151

e. PMID: 33323507

f. PMID: 35458509

We thank the reviewer for highlighting these works. We added a discussion item regarding tRNA expression in HCMV and other herpesviruses with the references (pages 15-16, lines 447-458).

10. CoV2 Discussion point-The lack of tRNA expression regulation might have more to do with the length of the infection (6 hpi cov2- also didn't see much a change at 5hpi with hcmv). This should be proposed as a possibility.

It is a possibility that due to the high stability of tRNAs, expression regulation of tRNAs will not affect the tRNA pool in short infection such as of SARS-CoV-2. We added this explanation in the discussion part, page 15, lines 441-442.

11. Line 582. Misspelled schlafen in Discussion. (SLFN, not SFLN)

The point is fixed in the revised manuscript.

Reviewer #3 (Significance (Required)):

General assessment: I found this paper exciting to read, given the dearth of knowledge regarding viral modulation of tRNA expression.

We appreciate the reviewer's comment

However, the work is highly descriptive, with a complete absence of follow-up or validation studies. At the very least, I would have hoped that the authors validated that viral titer (and not just GFP intensity) was impacted by some of the hits. The lack of confirmation and quality control overall diminishes confidence in the stated conclusions.

However, I think the topic is timely, important, and that this manuscript offers tools to the community at large to learn more about viral manipulation or other drivers of tRNA regulation. Once follow-up/validation experiments are added to the work, as detailed below, this manuscript will be of broad importance and highly impactful.

As mentioned above, we plan to add such validations to the fully revised manuscript.

Advance: While there have been many studies suggesting tRNA regulation occurs during viral infection (these pubs should be referenced as mentioned above), this is an advance due to the fact that it begins to address whether tRNA expression changes functionally impact viral replication. This will be much more solid with follow-up experiments confirming that hits alter HCMV replication (rather than GFP intensity).

Audience: This will be of broad interest to those with interest in virology and gene expression. The new sub-libraries of tRNA-related factors might be useful to be tested in other cell types

Revision Plan

and settings. Again, as the work stands, it is descriptive and hypothesis-stimulating, but the conclusions need validation and further support.

We thank the referee for the encouraging words and the suggested analyses. We already implemented most of the suggestions in the current revised version and hope to add further experiments in a fully revised manuscript.

9th Sep 2024

Manuscript Number: MSB-2024-12614-T

Title: Essentiality and dynamic expression of the human tRNA pool during viral infection

Author: Noa Aharon-Hefetz

Michal Schwartz

Orna Dahan

Noam Stern-Ginossar

Yitzhak Pilpel

Dear Dr. Pilpel,

Thank you for the submission of your manuscript to Molecular Systems Biology. I have now had a chance to carefully read your manuscript and revision plan. We think the study is of potential interest, and we would like to invite a major revision of your manuscript.

We think your revision plan appears reasonable. In particular, Reviewers #1 and #3 noted that the current study remains largely descriptive. Therefore, it would be crucial to perform experimental follow-up and validations as suggested by Reviewer #3 to strengthen the conclusion and enhance the overall impact of the study. Additionally, as mentioned by Reviewer #1, attention should be given to the presentation of the study, ensuring that the main conclusions are communicated more clearly and cohesively.

As you may already know, our editorial policy allows in principle a single round of major revision, and it is therefore essential to respond to the reviewers' comments that are as complete as possible. Please feel free to contact me in case you would like to discuss in further detail any of the issues raised by the reviewers.

On a more editorial level, we would ask you to address the following issues:

- Please provide a .docx formatted version of the manuscript text (including legends for main figures, EV figures and tables). Please make sure that the changes are highlighted to be clearly visible.

- Please provide individual production quality figure files as .eps, .tif, .jpg (one file per figure).

- Please provide a .docx formatted letter INCLUDING the reviewers' reports and your detailed point-by-point responses to their comments. As part of the EMBO Press transparent editorial process, the point-by-point response is part of the Review Process File (RPF), which will be published alongside your paper.

- Please note that all corresponding authors are required to supply an ORCID ID for their name upon submission of a revised manuscript.

- We replaced Supplementary Information with Expanded View (EV) Figures and Tables that are collapsible/expandable online (see examples in <http://msb.embopress.org/content/11/6/812>). A maximum of 5 EV Figures can be typeset. EV Figures should be cited as 'Figure EV1, Figure EV2' etc... in the text and their respective legends should be included in the main text after the legends of regular figures.

Additional Tables/Datasets should be labeled and referred to as Table EV1, Dataset EV1, etc. Legends have to be provided in a separate tab in case of .xls files. Alternatively, the legend can be supplied as a separate text file (README) and zipped together with the Table/Dataset file.

For the figures and tables that you do NOT wish to display as Expanded View figures, they should be bundled together with their legends in a single PDF file called *Appendix*, which should start with a short Table of Content. Each legend should be below the corresponding Figure/Table in the Appendix. Appendix figures and tables should be referred to in the main text as: "Appendix Figure S1, Appendix Figure S2, Appendix Table S1" etc. See detailed instructions regarding expanded view here: <https://www.embopress.org/page/journal/17444292/authorguide#expandedview>.

- Before submitting your revision, primary datasets (and computer code, where appropriate) produced in this study need to be deposited in an appropriate public database (see <http://msb.embopress.org/authorguide> - dataavailability <https://www.embopress.org/page/journal/17444292/authorguide#dataavailability>).

The accession numbers and database should be listed in a formal "Data Availability" section (placed after Materials & Method) that follows the model below (see also <https://www.embopress.org/page/journal/17444292/authorguide#dataavailability>). Please note that the Data Availability Section is restricted to new primary data that are part of this study.

Data availability

- RNA-Seq data: Gene Expression Omnibus GSE46843 (<https://www.ncbi.nlm.nih.gov/geo/query/acc.cgi?acc=GSE46843>)

- [data type]: [name of the resource] [accession number/identifier/doi] ([URL or identifiers.org/DATABASE:ACCESSION])

-At EMBO Press we ask authors to provide source data for the main figures. Our source data coordinator will contact you to discuss which figure panels we would need source data for and will also provide you with helpful tips on how to upload and organize the files.

- Our journal encourages inclusion of *data citations in the reference list* to directly cite datasets that were re-used and obtained from public databases. Data citations in the article text are distinct from normal bibliographical citations and should directly link to the database records from which the data can be accessed. In the main text, data citations are formatted as follows: "Data ref: Smith et al, 2001". In the Reference list, data citations must be labeled with "[DATASET]". A data reference must provide the database name, accession number/identifiers and a resolvable link to the landing page from which the data can be accessed at the end of the reference. Further instructions are available at .

- We updated our journal's competing interests policy in January 2022 and request authors to consider both actual and perceived competing interests. Please review the policy <https://www.embopress.org/competing-interests> and update your competing interests if necessary.

Please use the heading "Disclosure statement and competing interests".

- All Materials and Methods need to be described in the main text using our 'Structured Methods' format. According to this format, the Methods section includes a Reagents and Tools Table (listing key reagents, experimental models, software and relevant equipment and including their sources and relevant identifiers) followed by a Methods and Protocols section describing the methods, ideally using a step-by-step protocol format. The aim is to facilitate adoption of the methodologies across labs. Please download and fill our Reagents and Tools Table template (.docx), which you can find in our author guidelines: <https://www.embopress.org/page/journal/17444292/authorguide#structuredmethods>.

-Regarding data quantification:

Please ensure to specify the name of the statistical test used to generate error bars and P values, the number (n) of independent experiments (please specify technical or biological replicates) underlying each data point and the test used to calculate p-values in each figure legend. Discussion of statistical methodology can be reported in the materials and methods section, but figure legends should contain a basic description of n, P and the test applied.

Graphs must include a description of the bars and the error bars (s.d., s.e.m.).

- Please provide a "standfirst text" summarizing the study in one or two sentences (approximately 250 characters, including space), three to four "bullet points" highlighting the main findings and a "synopsis image" (550px width and 400-600 px height, PNG format) to highlight the paper on our homepage.

Here are a couple of examples:

<https://www.embopress.org/doi/10.15252/msb.20199356>

<https://www.embopress.org/doi/10.15252/msb.20209475>

<https://www.embopress.org/doi/10.15252/msb.209495>

When you resubmit your manuscript, please download our CHECKLIST (<https://www.embopress.org/pb-assets/embo-site/EMBO%20Press%20Author%20Checklist-1642513524327.xlsx>) and include the completed form in your submission.

Please note that the Author Checklist will be published alongside the paper as part of the transparent process (<https://www.embopress.org/page/journal/17444292/authorguide#transparentprocess>).

If you feel you can satisfactorily deal with these points and those listed by the referees, you may wish to submit a revised version of your manuscript. Please attach a covering letter giving details of the way in which you have handled each of the points raised by the referees. A revised manuscript will be once again subject to review and you probably understand that we can give you no guarantee at this stage that the eventual outcome will be favorable.

I look forward to receiving your revised manuscript soon.

Kind regards,
Jingyi

Jingyi Hou, PhD
Scientific Editor
Molecular Systems Biology

We realize that it is difficult to revise to a specific deadline. In the interest of protecting the conceptual advance provided by the work, we recommend a revision within 3 months (9th Oct 2024). Please discuss the revision progress ahead of this time with the editor if you require more time to complete the revisions. Use the link below to submit your revision:

IMPORTANT: When you send your revision, we will require the following items:

1. the manuscript text in LaTeX, RTF or MS Word format
2. a letter with a detailed description of the changes made in response to the referees. Please specify clearly the exact places in the text (pages and paragraphs) where each change has been made in response to each specific comment given
3. three to four 'bullet points' highlighting the main findings of your study
4. a short 'blurb' text summarizing in two sentences the study (max. 250 characters)
5. a 'thumbnail image' (550px width and max 400px height, Illustrator, PowerPoint or jpeg format), which can be used as 'visual title' for the synopsis section of your paper.
6. Please include an author contributions statement after the Acknowledgements section (see <https://www.embopress.org/page/journal/17444292/authorguide>)
7. Please complete the CHECKLIST available at (<https://bit.ly/EMBOPressAuthorChecklist>). Please note that the Author Checklist will be published alongside the paper as part of the transparent process (<https://www.embopress.org/page/journal/17444292/authorguide#transparentprocess>).
8. When assembling figures, please refer to our figure preparation guideline in order to ensure proper formatting and readability in print as well as on screen:
<https://bit.ly/EMBOPressFigurePreparationGuideline>
See also figure legend guidelines: <https://www.embopress.org/page/journal/17444292/authorguide#figureformat>
9. Please note that corresponding authors are required to supply an ORCID ID for their name upon submission of a revised manuscript (EMBO Press signed a joint statement to encourage ORCID adoption). (<https://www.embopress.org/page/journal/17444292/authorguide#editorialprocess>)
Currently, our records indicate that the ORCID for your account is 0000-0003-3200-9344.

Link Not Available

11. Include a Reagents and Tools Table as part of the Methods section, which can be downloaded from our author guidelines (<https://www.embopress.org/page/journal/17444292/authorguide#structuredmethods>)

*** PLEASE NOTE *** As part of the EMBO Press transparent editorial process initiative (see our Editorial at <https://dx.doi.org/10.1038/msb.2010.72>), Molecular Systems Biology publishes online a Review Process File with each accepted manuscripts. This file will be published in conjunction with your paper and will include the anonymous referee reports, your point-by-point response and all pertinent correspondence relating to the manuscript. If you do NOT want this File to be published, please inform the editorial office at msb@embo.org within 14 days upon receipt of the present letter.

Rev_Com_number: RC-2024-02487

New_manu_number: MSB-2024-12614-T

Corr_author: Pilpel

Title: Essentiality and dynamic expression of the human tRNA pool during viral infection

Attention Dr. M. Madan Babu (Chief Editor, *MSB*)

Dear Madan,

We hereby submit the revised version of our paper titled "Essentiality and dynamic expression of the human tRNA pool during viral infection". The paper was reviewed on a first round rather positively, although some useful comments, concerns, and advice came up. We have further corresponded with you with a detailed plan of experiments to be done. After getting a green light, we have carried out the requested experiments, analyses, and edits and addressed all points raised by the reviewers. We are getting good support for our original results and are confident that the manuscript has been improved and is now better grounded. We very much hope you and the referees will find it now suitable.

With best regards, Tzachi.

Reviewer #1 (Evidence, reproducibility and clarity (Required)):

I have mixed feelings regarding this manuscript. On the one hand, the authors did an impressive amount of work. On the other hand, the manuscript seems overly descriptive (writing should be more concise) without a clear message or hypothesis that is cohesive to all the presented evidence. Below, I will outline my concerns.

We appreciate the comment about missing a cohesive presentation. We worked extensively to improve that in the revised manuscript.

Reviewer #1- first part

1. I am not an expert in the field of viral biology and immunology. I wonder how well the IFN treatment mimics the cellular response to infection (yet without the virus). Also, how good is ruxolitinib at blocking the IFN response? I would appreciate it if you could explain both with one or two sentences and provide the necessary references.

The reviewer is correct that we cannot claim that interferon treatment mimics exactly the cellular response to a viral infection. However, the expression of interferon-stimulated genes (ISGs) is a major arm of the antiviral response to HCMV (c.f. [doi:10.3390/v10090447](https://doi.org/10.3390/v10090447), [doi:10.2217/fvl-2018-0189](https://doi.org/10.2217/fvl-2018-0189)). In addition, Ruxolitinib is a potent and selective Janus kinase 1 and 2 inhibitor ([doi:10.1021/ol900350k](https://doi.org/10.1021/ol900350k)), and we have shown in the past that it very effectively reduces the expression of many ISGs ([doi: 10.1038/s41590-018-0275-z](https://doi.org/10.1038/s41590-018-0275-z)). Since ISGs constitute a major part of the host response to HCMV infection, the fact that their expression leads to minor changes in the tRNA pool strongly suggests that mainly the virus mediates the changes seen in the tRNA pools during HCMV infection. In the revised version, these claims were amended, and relevant references were added (pages 5, lines 116-118).

2. (MAJOR) Can these two treatments really allow the effects of host response and viral

infection to be separated? OR in other words, are these two effects really orthogonal? In my opinion, they are NOT. Fig. 1E seems to support my opinion, as the changes seen for the "IFN" sample relative to the "uninfected" sample (referred to as "changes-A" below), are parallel to the changes seen for the "24hpi + ruxo" sample relative to the "24hpi" sample ("changes-B"). More specifically, changes-A represent the host response, as argued by the author, whereas changes-B represent the elimination of the host response (due to ruxo, conditioned on the virus-driven effect). If the virus-driven effect and the host response could really be separated, one would expect changes-A and changes-B are more or less opposite. However, they appeared to be parallel, suggesting that uninfected versus infected conditions can have totally different (even opposite) host responses. More importantly, if one cannot separate the host response from virus-driven effects, the conclusion of "tRNA changes are driven by virus, not host response" is then unfounded.

This is an important point to clarify. We used complementary perturbations (IFN and ruxolitinib treatments) to infer their relative contributions and not to claim complete separation. Therefore, we cannot discuss the directionality of the host and viral effects, and we removed this point from the revised manuscript. However, the PCA (Figure 1E) and the hierarchical clustering (Figure 1F) do suggest that the viral effect on the tRNA pool is more dominant than the host effect. In the PCA, the biggest change between uninfected samples to infected samples is along the 1st PC, with 62% variance explained (Figure 1E). From the heatmap in Figure 1F, we see that the tRNA pool is almost unchanged in the IFN sample relative to uninfected, and the profound changes in the tRNAs are seen in infected samples, and more so in the ruxo-treated sample. This point was clarified in the revised version (page 5, lines 133-136).

3. Even if we let go of this previous point and accept that these results indeed offer some support for the notion that the virus-driven effect are the main contributor to the shifts in tRNA pool, the support is at best moderate. A big gap here is "how?" I suggest the authors should at least give some insight on how virus can do that in Discussion (and mention it with one sentence in Results).

We certainly welcome the challenge, which we now meet in the revision. In short, transcription regulation of tRNAs, mainly upon viral infection, is poorly studied. Herpesviruses like HSV-1, MHV68, and EBV were shown to increase tRNA expression through manipulation of RNA polymerase III ([10.1128/mBio.02664-20](https://doi.org/10.1128/mBio.02664-20), [10.1038/s41467-022-28144-8](https://doi.org/10.1038/s41467-022-28144-8), doi.org/10.1016/j.semcd.2023.01.011). A similar effect was also shown upon HCMV infection, which regulates the entire host transcription machinery ([10.3390/v14040779](https://doi.org/10.3390/v14040779)). However, which HCMV factors promote upregulation in the tRNA transcription machinery is still unknown. In MHV68, at least three viral proteins (ORF36, ORF45, ORF37/muSOX) induce pre-tRNA accumulation ([10.1128/mBio.02664-20](https://doi.org/10.1128/mBio.02664-20), [10.1128/jvi.00262-20](https://doi.org/10.1128/jvi.00262-20), [10.1128/spectrum.00172-23](https://doi.org/10.1128/spectrum.00172-23)). We now relate to this point in the results (page 5, lines 135-136) and discuss the known effects of viral infection on tRNA expression in the discussion section (page 13, lines 376-388).

4. The authors compared the HCMV codon usage to the proliferation and differentiation

signatures of human cells. But these two signatures are not compared with measured tRNA expression. It might shed some light on the general characteristics of tRNA pool shifts due to infection (towards a proliferation-like or differentiation-like signature). This fits in the general topic of virus-host interaction and might give more evidence for the point that HCMV is adapted to a differentiation signature (as it drives the host into that state).

We thank the reviewer for the suggestion. We performed the requested analysis (Figure 1). We found that the tRNA pool of uninfected HFF cells correlated to the same extent with proliferation codon usage ($r=0.29$, $p\text{-value}=0.029$) and differentiation codon usage ($r=0.26$, $p\text{-value}=0.05$). Similar correlations to the proliferation and differentiation signatures were found when analyzing the tRNA pool 72h post-infection (proliferation $r=0.33$, $p\text{-value}=0.011$, differentiation $r=0.28$, $p\text{-value}=0.034$). This result suggests no general shift in the tRNA pool towards a specific codon usage signature.

Figure 1: Correlations between differentiation (upper panel) or proliferation (lower panel) codon usage signatures to the tRNA pool of uninfected HFF (left panel) or 72hpi (right panel). Each dot corresponds to one of the 56 codons with a corresponding tRNA that was detected in the tRNA sequencing. The x-axis depicts codon usage of proliferation or differentiation signatures, and the y-axis the tRNA level (\log_2) in uninfected or 72hpi HFF.

5. How is the dashed box in Fig3A/B chosen?

We determined the dashed lines based on the most prominent groups of transcripts that are best adapted to proliferation (18%) or differentiation (30%) codon usage signatures. Appendix Figure S3A clearly shows the two groups without viral genes. We emphasize this point in the legend of Appendix Figure S3A.

6. The tAI values shown in Fig3C-E are extremely low (compared to other reports I am aware of). Does this mean that the adaptation of viral codon usage to human cell supply is actually very weak? This is in opposition to the major claims made in this section.

We acknowledge that the tAI values presented here are lower than typically presented. However, this is due to how tAI was calculated rather than the potential weak adaptation

between viral genes and tRNA supply. Specifically, unlike previous works that estimate tRNA availability based on gene copy number, here we calculated tAI using tRNA sequencing (to capture the dynamics in the tRNA pool during infection). Indeed, the value of tAI calculated by tRNA read counts is lower than tAI calculated by tRNA copy number. This is due to the skewed distribution of tRNA read counts (many tRNAs are highly expressed, and a few are lowly expressed), while tRNA copy number is distributed more evenly. Thus, due to the mathematical nature of the tAI (computing geometric rather than arithmetic average of tRNA availability), the skewed distribution observed in the data results in lower tAI values. Nevertheless, as all tAI calculations here were done similarly, the comparisons between gene groups or genes are valid. We also compute tAI based on tRNA gene copy number for comparison, and show higher and more typical values (Figure 2).

Figure 2- A violin plot describing the tAI calculated for all canonical HCMV genes based on the gene copy number of the human tRNA pool.

7. I believe that the part about SARS-CoV-2 could be made more concise. It is sufficient to mention that results may differ from those obtained with HCMV in one paragraph.

We appreciate the comment and have revised the text accordingly. In the revised manuscript, the section on SARS-CoV-2 has been shortened significantly, emphasizing only that results may differ from those obtained with HCMV.

8. Line 299 on page 11 - I do not believe codon usage between different viruses can be directly compared, let alone reaching such a conclusion. Some viruses have low CAI or tAI to humans, but they have co-evolved with humans for a long time. Furthermore, there are viruses that infect multiple hosts, but their CAI for a host with which they have long co-evolved is higher while their CAI for a host that is relatively new is lower.

We agree with the reviewer that a direct link between co-evolution time and tAI may not always exist. Indeed, other factors might explain the observation that SARS-CoV-2 genes are

less adapted than HCMV genes. These may include effective population sizes and mutation rates that vary substantially between the viruses. We, therefore, removed this statement from the manuscript.

9. (MAJOR) A more general comment is that there is a difference between tRNA expression and the abundance of translation-ready tRNA. The process of charging tRNA with amino acids may take a long time. It is the abundance of the charged-tRNA (the ternary complex of aminoacylated tRNA and EF-Tu-GTP) that is of biological importance. In this regard, the use of tRNA expression falls short.

The reviewer raises a valid point. Indeed, our tRNA sequencing protocol measures both charged and uncharged tRNAs that constitute the cell's mature tRNA pool. Compared to previous studies ([10.1128/mbio.02664-20](https://doi.org/10.1128/mbio.02664-20), [10.3390/v14040779](https://doi.org/10.3390/v14040779)) that focus on the transcription process of tRNAs in viral-infection models by sequencing the pre-tRNAs, here we look at the mature tRNA pool that accounts for both transcription and post-transcription processes. Therefore, we changed the mention of "tRNA expression" to "mature-tRNA levels" in the manuscript. We note, however, that although limited in the ability to differentiate between charged and uncharged tRNAs, the tRNA sequencing protocol used here is commonly used and validated as a state-of-the-art protocol in tRNA sequencing ([10.1016/j.molcel.2021.01.028](https://doi.org/10.1016/j.molcel.2021.01.028), [10.1038/s41467-020-17879-x](https://doi.org/10.1038/s41467-020-17879-x), etc.), mainly because it addresses the level of "ready-to-use" tRNA. We note this point in the Discussion, page 13, lines 376-378.

Reviewer #1- second part

1. (MAJOR) Prior to the actual competition assay in the first high-throughput screen (cell competition assay), the authors applied two days of antibiotic selection and two days of recovery. This could result in a serious problem of false negatives or drop outs. Specifically, an sgRNA targeting an essential gene with high efficiency would kill the cells, leaving no (or a small number of) cells in the ancestor population at the beginning of the competition process. A sgRNA's enrichment in competing populations cannot be reliably estimated in such situations. I am not certain that the FDR used in Figure 5B is sufficient to address this issue. Please clarify whether it could. Providing raw counts for competing and ancestor populations would also be helpful.

As customary in CRISPR screens, the step of lentiviral transduction and antibiotic selection is necessary to ensure that only CRISPR-edited cells are left in the population ([10.1016/j.xpro.2023.102201](https://doi.org/10.1016/j.xpro.2023.102201)). Indeed, essential genes like housekeeping genes might be removed from the competing population relatively quickly, resulting in their dropouts. We could have lost some tRNA hits in the cell growth CRISPR screen (Figure 5B-C) because of their overall essentiality for cell growth. The MAGcK tool we used, the state-of-the-art in the field, filters out sgRNAs with low read counts to be able to calculate false discovery rates, thus dropouts are removed from the analysis. Reassuringly, we identified 15 tRNAs that were depleted from the population during competition even after removing dropouts. Therefore, while every CRISPR screen such as this might be subject to false negatives, we certainly have a substantial amount of true positive discoveries. tRNA dropout in the HCMV infection

CRISPR screen (Figure 6B-C) can also happen, which means our screen underestimates the essentiality of tRNAs to HCMV infection. However, this concern does not affect the significance of the hits we did find. We acknowledge this inherent difficulty in CRISPR screens, and we provided the raw read counts of all samples in GEO as instructed (X). We do emphasize, though, that while valid, this concern applies to essentially any CRISPR screen that is commonplace in genomics these days.

2. It is also highly questionable to me the nearly negligible effects of tRNA modification enzymes. This may be explained by the point above. Indeed, the dots of tRNA modification enzymes in general appear to have higher FDR (lower y values) when compared to red dots with similar enrichment levels.

This is a valid point. We found a lack of essentiality of tRNA modification enzymes in both screens. Although potentially surprising, the lack of essentiality of tRNA modification enzymes is observed in a different HCMV-CRISPR screen ([10.1038/s41587-021-01059-3](https://www.ncbi.nlm.nih.gov/geo/query/acc.cgi?acc=GSE110593)) and analyzed in the revised manuscript (Appendix Figure S5C). One potential reason is that some of these enzymes were "backed up" by others, which we now mention (page 15, lines 424-425). Another explanation is that most tRNA modification enzymes are indeed not essential for growth and viral infection (now described in the Discussion, page 15, line 424). Alternatively, dropouts can explain this result, as suggested by the reviewer (page 15, line 423). To examine the likelihood of the dropout scenario, we examined the average raw read count of the sgRNAs targeting the tRNA modification enzymes in the ancestor samples. We compared it to that of other protein-coding and non-targeting sub-groups. We found that the sgRNAs targeting tRNA modification enzymes received similar read counts to other sub-groups in the CRISPR library (Figure 3). Thus, we now explain (page 10, lines 267-271) that the dropout issue is not likely to explain our screens' lack of essentiality of tRNA modification enzymes.

Figure 3- histogram describing the raw read count of all sgRNAs targeting tRNA modification enzymes. The dashed lines describe the average read counts of sgRNAs targeting other subgroups (NT -non-targeting (gray), control genes - growth dependency factors (blue), growth restriction factors (red), HCMV-infection dependency factors (green), and HCMV-infection restriction factors (purple)).

3. The screen based on IE2-GFP labeled HCMV measures a phenotype that is very difficult to interpret. Particularly, I am not sure if GFP2 and GFP3 are good controls for comparing GFP4 (GFP1 might be better). Various factors can affect GFP levels, including, but not limited to, dilution caused by a rapidly dividing host cell, unhealthy translational machinery resulting from infection or microenvironment. My point is supported by some observations in Fig6B.

For example, SEC61B, a restriction factor for HCMV infection, is enriched in the GFP2 group, contrary to expectations. It is necessary for the authors to prove with firm evidence that their choice of GFP signal thresholds is appropriate.

We acknowledge the concern. Specifically, the translation of the GFP gene itself could be thought to be affected by the tRNA manipulation. To account for this potential concern, we tested the codon usage (a proxy for demand for each tRNA) of the eGFP gene (which is the GFP version we used in the system) and compared it with tRNA essentiality, as determined by the cell growth CRISPR screen. Note that codons with perfect or imperfect assignment to their corresponding tRNAs (due to wobble) received the same essentiality values for their tRNA. Figure 4 below shows that GFP does not tend to significantly use codons that correspond to essential or less essential tRNAs. We report this in the revised manuscript (page 11, lines 316-318, and added Appendix Figure S6D). The same lack of correlation was also found for the tRNA essentiality upon HCMV infection (not shown)

Figure 4: Each dot corresponds to one of the 61 codons in the genetic code. The x-axis depicts codon usage in GFP, and the y-axis the essentiality of the corresponding tRNA in the CRISPR screen

More generally, we show that GFP intensity does correlate with viral genome copies (Appendix Figure S6A). Also, analysing mRNA-seq data of temporal HCMV infection ([10.1016/j.celrep.2022.110653](https://doi.org/10.1016/j.celrep.2022.110653)) reveals that IE2 (UL122) shows a dynamic expression- a high expression peak in early infection, then a decline in expression level followed by a gradual increase (Figure 5).

Figure 5: Dynamic expression of UL122 (IE2) during HCMV infection. The X-axis depicts the time point along the HCMV infection, and the Y-axis depicts the normalized mRNA expression of UL122.

Altogether, we believe that the IE2-GFP level provides a good estimation for viral load ([10.1038/s41467-024-45614-3](https://doi.org/10.1038/s41467-024-45614-3), [10.1128/mbio.00337-22](https://doi.org/10.1128/mbio.00337-22)).

Regarding SEC61B, which served as a control in our screen, the referee is rightly asking why it behaves oppositely from what's expected, given that this was supposed to be a restriction factor of HCMV infection. We returned to the literature on the essentiality of this gene upon HCMV infection. In Weissman's paper ([10.1038/384432a0](https://doi.org/10.1038/384432a0)), which was the reference for choosing control genes in our system, this gene was targeted through two different CRISPR technologies, once with CRISPR knockout and once with CRISPRi. Interestingly, only upon CRISPRi did this gene prove to be a restriction factor (i.e., improved infection upon reduction of the gene). We comment on this peculiar fact in the revised manuscript (page 11, lines 301-304). However, we note that the rest of our positive and negative controls deliver the expected results – increasing or reducing infection upon KO, as expected from their role as restriction or dependency factors, respectively, thus lending considerable support to our experimental system. It is possible, especially in light of our screen, and since other positive and negative controls behave as expected, that the status of this gene as a "restriction factor" of viral infection needs to be reconsidered.

I would appreciate more information regarding why restriction factors of cell growth have a high GFP2/GFP4. Intuitively, a KO of restriction factors of cell growth should result in better growth and higher GFP, thus leading to enrichment in GFP4, not GFP2.

The reviewer raises an interesting question (although not at the heart of this work, as sgRNAs for the cell growth restriction factor mainly aim to serve as controls for the CRISPR screen). HCMV has a complex interaction with the cellular cell cycle. Specifically, it establishes a unique G1/S arrest that is both stimulatory and inhibitory. On the one hand, it serves the virus for the cell cycle arrest, a critical step for viral genome replication. On the other hand, the virus needs many resources that also serve cell growth. Both p53 and CDKN1A (p21) are important regulators at this stage. Also, p53 levels are stabilized during HCMV infection. ([doi: 10.1128/mBio.02934-21](https://doi.org/10.1128/mBio.02934-21), [10.1128/jvi.76.11.5369-5379.2002](https://doi.org/10.1128/jvi.76.11.5369-5379.2002), [doi: 10.1128/jvi.00505-06](https://doi.org/10.1128/jvi.00505-06)). Therefore, it is not necessarily surprising that genes related to cell growth and the cell cycle act as dependency factors for HCMV infection. We now discuss this issue in page 11, lines 304-308.

5. Line 404 "nonetheless"

We appreciate the reviewer for noticing the typo. We corrected it.

Reviewer #1 (Significance (Required)):

The relation between human tRNA supply and viral translation is a topic of profound biological and biomedical importance. In this study, the authors used HCMV infection as the primary model to investigate this question. Results fall into two major parts: (i) changes in the tRNA pool during viral infection, and (ii) the impact of tRNA-related gene KO on viral infection.

We appreciate the detailed report. We addressed all points raised in the revised manuscript.

Reviewer #2 (Evidence, reproducibility and clarity (Required)):

In this study by Aharon-Hefetz et al., the researchers examined changes in tRNA pools during virus infections. The translation machinery plays a crucial role in virus replication.

Consequently, host cells have developed sensors and effectors within this compartment to counteract viral mechanisms. The translation apparatus serves as a pivotal point in the virus-host conflict. Therefore, investigating alterations in the translation machinery during infections is vital for gaining a comprehensive understanding of the infection process.

This study offers a thorough and high-quality analysis of data in a relevant cell culture system involving two different viruses. By conducting tRNA sequencing, the researchers studied the human tRNA pool following infections with human Cytomegalovirus (HCMV) and SARS-CoV-2. Changes in tRNA expression induced by HCMV were mainly driven by the virus infection itself, with minimal impact from the cellular immune response. Interestingly, specific tRNA post-transcriptional modifications seemed to influence stability and were subject to manipulation by HCMV. Conversely, SARS-CoV-2 did not lead to significant alterations in tRNA expression or post-transcriptional modifications.

Moreover, a systematic CRISPR screen targeting human tRNA genes and modification enzymes allowed the identification of specific tRNAs and enzymes that either enhanced or reduced HCMV infectivity and cellular growth. This information enabled them to control the development of HCMV-specific tRNA modifications, highlighting the importance of these tRNA epitranscriptome modifications in virus replication.

The authors concluded that the observed differences between the viruses are consistent with HCMV genes aligning with differentiation codon usage and SARS-CoV-2 genes reflecting proliferation codon usage. This observation's connection to the biology of HCMV and SARS-CoV-2 lies in the codon usage of structural and gene expression-related viral genes, showing a significant adaptation to host cell tRNA pools. Notably, these genes from both viruses demonstrated the highest adaptation to the tRNA pool of infected cells. The reason behind this phenomenon remains unclear. One hypothesis suggests that a high level of structural gene expression is necessary during activation. Testing this hypothesis could involve examining if hindering tRNA modifications affects virus morphogenesis.

In summary, this study presents an interesting and innovative perspective on how viruses modify the translation machinery. The meticulous analysis sheds light on a central interaction point between viruses and their host cells.

Reviewer #2 (Significance (Required)):

In summary, this study presents an interesting and innovative perspective on how viruses modify the translation machinery. The meticulous analysis sheds light on a central interaction point between viruses and their host cells.

We thank the reviewer for finding our work interesting, innovative, and well analyzed.

Reviewer #3 (Evidence, reproducibility and clarity (Required)):

Summary

Aharon-Hefetz et al. present the expression dynamics and modification signatures of tRNAs using DM-tRNA-seq in human foreskin fibroblasts or Calu3 cells during infections with two diverse viruses, HCMV and SARS-CoV2, respectively. They also use a newly designed tRNA-centric CRISPR library to screen the essentiality of tRNA and tRNA factors during HCMV-GFP infection. They find several tRNAs that are differentially expressed during HCMV infection, and most closely resemble the set of tRNAs shown to be used during cellular differentiation. Additionally, tRNA differential expression does not resemble that following interferon treatment, implying that virus modulation of tRNAs is unique to the general interferon response. They compare codon usage signatures during infection to their prior-defined sets of proliferation/differentiation tRNA genes. In their CRISPR screen, they find that different tRNAs can promote or restrict HCMV infection levels, as measured by the intensity of GFP fluorescence marker in their virus. Surprisingly, there were few tRNA modification factor hits that contributed to growth or infection.

Reviewer #3- major comments

1. The topic of this work is important, and the analysis performed here is assumed to be top quality, based on the previous work by the last author. The weakness with this body of work is a lack of rigor, specifically regarding validation and follow-up studies. Without these experiments, the reader lacks confidence in stated conclusions. For example:
 - a. There is no validation or clue to how penetrant CRISPR is against tRNA genes. Given how duplicated some tRNA families are, it is possible that CRISPR is more effective against certain families compared to others. While this is likely an inherent caveat in all CRISPR screens, it would lend confidence in this approach to see some validation of tRNA KO by northern blot or RT-qPCR or sequencing.

We thank the reviewer for raising this important issue. Indeed, many tRNA genes appear in multiple copies in the human genome. Yet, based on our previous work, we expect parallel editing of multiple copies using the same sgRNA. In our previous work (doi.org/10.7554/eLife.58461), we validated, based on several tRNA families, the ability of our tRNA CRISPR system to successfully target and affect tRNA expression levels. This included sequencing of the edited tRNA genes (i.e., DNA sequencing), in which we observed diverse INDEL mutations that predicted full disruption of the tRNA structure. Furthermore, we sequenced the CRISPR-edited cells' tRNA pool and found the targeted tRNAs' downregulation to be up to 2- 4-fold. The previous work provides confidence in our tRNA-CRISPR approach.

Nevertheless, to further mitigate the reviewer's concern, we performed an additional experiment in which we individually knocked out our top tRNA hits (iMet-CAT and His-GTG knockout) and performed tRNA sequencing on those cells. We found a reduction of 4-fold in the iMet-CAT gene and ~1.4-fold in the His-GTG gene (relative to these genes' level in the non-targeting sample). Although the expression change in the His-GTG gene is modest, reassuringly, we see a corresponding significant functional effect in reducing

HCMV infection. This result indicates the efficiency of our designed sgRNAs in manipulating the targeted tRNA gene levels and activity. These results are now shown in Figure 6E-F, and in the Results part (page 11, lines 322-323).

b. There is no validation that tRNA modification factor knockouts alter tRNA modification levels. Without this knowledge, the lack of essentiality cannot be confidently and fully interpreted. If the group does not validate whether individual tRNA modification factor knockouts alter modification profiles, then all possible explanations should be posited. For example, it is possible that 1) there could be major redundancy among tRNA modification enzymes, as the authors posit in the Discussion 2) tRNA modification enzymes are not essential for growth bc their activity/the modification they place is non-essential for growth, OR 3) the knockouts are not fully penetrant. I think this Discussion should be expanded to make caveats clearer. Perhaps referencing whether tRNA modification factors have been shown to be essential in other CRISPR screens would be helpful.

Thank you for raising this important point. Guided by your advice, we chose two tRNA modification enzymes from our CRISPR library (FTSJ1 and TRMT112) and tried to knock them out individually using the best-performing sgRNAs from our library. We tested the effect of these knockouts on HCMV infection and found no significant difference relative to the non-targeting sgRNA control.

To further address this point, we also scanned the literature, looking for studies testing the effect of knocking out tRNA modification enzymes on modification levels and cells' phenotypes. We found mixed results. In some studies, knockout of tRNA modification enzymes resulted in a reduction in modification level and in cell growth ([10.1073/pnas.2106556118](https://doi.org/10.1073/pnas.2106556118), [10.1038/s41467-024-52389-0](https://doi.org/10.1038/s41467-024-52389-0)), while in other studies, knockout of tRNA modification enzymes showed no, minimal, or only partial effect in cell growth of different cell types ([10.1080/15476286.2020.1712544](https://doi.org/10.1080/15476286.2020.1712544), [10.1016/j.molcel.2021.06.031](https://doi.org/10.1016/j.molcel.2021.06.031)). It seems that our original and current experimental results are certainly within the range of observed effects of tRNA modification enzymes knockouts.

Our results raise certain explanations, as was suggested by the reviewer. We now acknowledge these possibilities (e.g., insufficient targeting of the CRISPR system to target tRNA modification enzymes, limited role of the modification in cell growth or HCMV infection, and functional redundancy among those genes) in the Discussion (page 15, lines 423-425).

c. There is no validation that factors modulating GFP intensity in the HCMV screen actually impact virus replication. This is the point most important to this body of work. While GFP intensity does correlate to genome copies as shown by the authors, GFP read-out on a case-by-case basis could be simply due to factors required for expression/translation of GFP. Are any of the tRNA hits enriched or not represented in GFP reporter sequence? Either way, this information is informative.

We acknowledge the concern. Specifically, the translation of the GFP gene itself could be thought to be affected by the tRNA manipulation. To account for this potential concern, we tested the codon usage (a proxy for demand for each tRNA) of the eGFP gene (which is the GFP version we used in the system) and compared it with tRNA essentiality, as determined by the cell growth CRISPR screen. Note that codons with perfect or imperfect assignment to their corresponding tRNAs (due to wobble) received the same essentiality values for their tRNA. Figure 6 below shows that GFP does not tend to significantly use codons that correspond to essential or less essential tRNAs. We report this in the revised manuscript (page 11, lines 316-318, and added Appendix Figure S6D). The same lack of correlation was also found for the tRNA essentiality upon HCMV infection (not shown).

Figure 6: Each dot corresponds to one of the 61 codons in the genetic code. The x-axis depicts codon usage in GFP, and the y-axis the essentiality of the corresponding tRNA in the CRISPR screen.

Additionally, given that the hits are cross-compared ONLY to other infected (low intensity "GFP+") cells, and not to an uninfected population, there is no guarantee that these primarily drive HCMV infection. The top hits should be validated in HFFs, infected with HCMV, with resulting titers/viral gene expression/genome copies measured. Additionally, the reasons for not using a GFP- population as a control should be clarified.

We agree that additional experiments on some hits are warranted. We tested individual dependency and restriction tRNA hits in a single sgRNA knockout. We also included some established non-tRNA genes. We chose the best-performing sgRNAs from the CRISPR library for the individual CRISPR knockout. Following the gene knockout, we infected the cells with the HCMV merlin strain. At 72 hpi, we harvested the cells and measured the relative copy number of viral genomes by qPCR of the known HCMV gene UL44 (relative to a reference human gene B2M). We found a significant decrease (5 and 20-fold) of the viral genome copy number following His-GTG and iMet-CAT tRNAs KO relative to a non-targeting sgRNA sample. We also validated a decrease in their gene levels following the knockout using tRNA sequencing. These results are now shown in Figure 6D-F. We note that the knockout of restriction tRNAs didn't significantly affect the viral infection. We also show that knocking out PDGFRA, the cell receptor through which HCMV virions enter their host cells, reduced viral infection, and knocking out IFNAR2, an interferon receptor that is part of the viral stress response, didn't show a significant effect on viral infection. We note that it is known in CRISPR experiments that restriction

hits are harder to reproduce in single-sgRNA knockout assays, reflecting the context-specific nature of restriction effects and their dependence on competitive population dynamics(doi.org/10.1038/srep31782, doi.org/10.1186/s12915-023-01536-y). These experiments are now described in pages 11-12, lines 319-334.

The reviewers also suggest comparing GFP1/2/3 to an ancestor in addition to comparing them to GFP4. Towards that, we now show a GFP2 vs ancestor comparison (shown below, Figure 7). The results look very similar and are now added to the supplemental material of the revised manuscript (page 11, lines 313-314, Appendix Figure S6B).

Figure 7: A volcano plot for targeted gene hits from tRNA-CRISPR screen in HCMV infection. The x-axis shows the Z-score of log2 FC between lowly-infected cells (GFP2) and the ancestor (uninfected) population. The y-axis shows the $-\log_{10}$ FDR as calculated from MAGeCK. The genes are marked according to the sub-libraries. Significance is determined by $FDR < 0.05$. All values are calculated for three biological repeats.

2. Though careful codon usage analysis for HCMV versus the human host was analyzed, it seems pertinent to analyze whether the differentially expressed tRNAs during infection correlate to either codon usage profiles. Figure 3C and S3C intend to address this point for viral gene groups; however, I would encourage the authors to expand the description of these results to make them easier to interpret, especially for those not in the tRNA field. For example, "tRNA adaptation index (tAI)" is not defined in the text, but simply referenced. For clarity, you should include a brief explanation of what this measure describes. Following, when reporting results from Figure 3, the results can then be delivered with more specific and interpretable language. These steps will ensure maximal scientific communication to the audience.

We appreciate the reviewer's comment regarding the importance of scientific communication and making this manuscript easier to interpret, especially for readers unfamiliar with the world of tRNAs and translation efficiency. We added a description of our motivation to use tAI and the meaning of the measurement (page 8, lines 203-205). We also elaborated on the results part and made the results more interpretable (page 8, lines 206-207 and 211-212).

Finally, given that changes are most visible at 72 hpi, the analysis should include expression based on this time point for comparison.

Regarding the time point used for tAI calculation (Figure 3), we tested the tAI measured by the tRNA pool at 72hpi and got very similar results to those obtained using the tRNA pool measured at 24hpi. As 24hpi represents the peak of HCMV infection, we decided to present this analysis. In the revised version, we also added the analysis done using the tRNA pool measured 72hpi as suggested by the reviewer (Appendix Figure S3E).

Reviewer #3- minor comments

1. I would recommend more care in terminology used for the CRISPR screen (Figures 5 and 6) to make the manuscript easier to digest. Labeling sgRNAs-containing cells as "Reduced Growth/Infection" or "Increased Growth/Infection" is not immediately easy to understand. For example, saying this sgRNA "increased growth" could refer to the knockdown increasing growth OR could mean that this sgRNA was enriched in cells with increased growth, which are opposing. It might be more clear to state to use depleted/enriched terminology in these figure labels. This also applies to the text, be sure to plainly describe the terminology and what it means each time you refer to the CRISPR results.

This is a good point. Indeed, focusing on the significant enrichment of the sgRNAs, rather than their effect on growth or infection, is more straightforward. We changed the terminology accordingly in Figures 5C and 6C and the text in the current version.

2. Is there actual evidence that the new tRNA sgRNA library is more effective than that used previously? State if so.

To our knowledge, we are the only ones to generate a tRNA sgRNA library in higher eukaryotes. We assume the referee refers to our previous paper on the smaller-scale library (doi.org/10.7554/eLife.58461). The addition here is that the library is much more comprehensive (the previous one targeted only 20 tRNAs), and we improved the sgRNA design algorithm, especially avoiding off-targets between tRNA families. We point it out in the revised manuscript (page 14, lines 408-409).

3. Fig 1A-C: The cutoff for "red" symbol distinction is not stringent enough. 1.05 would be red, but that is not convincingly upregulated. The cutoff should be at least $FC > 1.2$.

We thank the reviewer for bringing our attention to this point. In the current version, we changed the cutoff of absolute fold change higher than 1.2 in Figures 1A-C and Appendix Figure S1A (also in legend).

4. Need thorough description of tRNA bioinformatics and modification analysis (citing past work is not appropriate here-need to make accessible to your audience).

We agree with the comment. Further thorough descriptions of tRNA bioinformatics and modification analysis are added in the revised version (page 5, lines 124-125, page 6, lines 149-152).

5. Line 182- Result headings could be more informative, even with small adjustments. For example "Specific tRNA modifications are modulated in response to HCMV infection" is more clear and accurate, as there are only a few measurable changes in tRNA modification. Limitations of using sequencing techniques to analyze modifications (versus MS) should also be discussed.

We changed the title accordingly. We also point to the limitations in tRNA modification detection using tRNA sequencing (page 6, lines 154-156)

6. It is not immediately clear why the viral plot looks different in Fig S3B compared to Fig 3B.

We thank the referee for spotting this. We employed different length cutoffs on the genes in each panel and fixed them in the revised manuscript.

7. Line 254-255. This point is not immediately clear-please include more specific language detailing the logic leading to this conclusion.

Indeed, the logic here was missing in our original submission. The idea was that longer genes are associated with gene conservation, hence functionality. Thus, non-canonical HCMV genes that are both long and codon-optimized might have a function during HCMV infection. We added this explanation to the text (page 7, lines 199-201).

8. Line 408- "may be essential"-I would modify the language here. Especially given there is no true comparison with uninfected cells.

We improved the language throughout the revised manuscript.

9. There are a number of recent publications profiling tRNA expression in herpesviruses. These should be mentioned and discussed in the context of this work. I know some were included in the reference list, but the body of work as a whole, and how this work fits in and pushes the horizon, could be further emphasized. It is quite impressive that this is a conserved feature of herpesvirus infection.

a. PMID: 36752632

b. PMID: 35110532

c. PMID: 34535641

d. PMID: 33986151

e. PMID: 33323507

f. PMID: 35458509

We thank the reviewer for highlighting these works. We added a discussion item regarding tRNA expression in HCMV and other herpesviruses with the references (pages 13, lines 376-388)

10. CoV2 Discussion point-The lack of tRNA expression regulation might have more to do with the length of the infection (6 hpi cov2- also didn't see much a change at 5hpi with hcmv). This should be proposed as a possibility.

It is a possibility that due to the high stability of tRNAs, expression regulation of tRNAs will not affect the tRNA pool in short infections such as SARS-CoV-2. We added this explanation in the discussion part, page 13, lines 370-371.

11. Line 582. Misspelled schlafen in Discussion. (SLFN, not SFLN)

The point is fixed in the revised manuscript.

Reviewer #3 (Significance (Required)):

General assessment: I found this paper exciting to read, given the dearth of knowledge regarding viral modulation of tRNA expression.

We appreciate the reviewer's comment.

However, the work is highly descriptive, with a complete absence of follow-up or validation studies. At the very least, I would have hoped that the authors validated that viral titer (and not just GFP intensity) was impacted by some of the hits. The lack of confirmation and quality control overall diminishes confidence in the stated conclusions.

However, I think the topic is timely, important, and that this manuscript offers tools to the community at large to learn more about viral manipulation or other drivers of tRNA regulation. Once follow-up/validation experiments are added to the work, as detailed below, this manuscript will be of broad importance and highly impactful.

As mentioned above, we added validations to the fully revised manuscript. WE believe that with added experiments, the paper is now better substantiated and validated.

Advance: While there have been many studies suggesting tRNA regulation occurs during viral infection (these pubs should be referenced as mentioned above), this is an advance due to the fact that it begins to address whether tRNA expression changes functionally impact viral replication. This will be much more solid with follow-up experiments

confirming that hits alter HCMV replication (rather than GFP intensity).

Audience: This will be of broad interest to those with interest in virology and gene expression. The new sub-libraries of tRNA-related factors might be useful to be tested in other cell types and settings. Again, as the work stands, it is descriptive and hypothesis-stimulating, but the conclusions need validation and further support.

We thank the referee for the encouraging words and the suggested analyses. We implemented the suggestions in the revised version and improved it significantly.

22nd Oct 2025

Manuscript Number: MSB-2024-12614R

Title: Essentiality and dynamic expression of the human tRNA pool during viral infection

Author: Noa Aharon-Hefetz

Michal Schwartz

Einav Aharon

Noam Stern-Ginossar

Orna Dahan

Yitzhak Pilpel

Dear Dr. Pilpel,

Thank you for sending us your revised manuscript. I apologize for the delay in getting back to you-I've been traveling for conferences and have only recently returned to my regular schedule. We have now heard back from the two reviewers who agreed to evaluate your revised study. As you will see below, both reviewers are overall satisfied with the performed revisions and supportive of publication.

Before we can proceed with formal acceptance, we kindly ask you to address the following remaining issues:

1. The remaining relatively minor issues raised by from Reviewers #1 and #2 (the original Reviewer #3).

On a more editorial level, please do the following:

1. Please provide up to five keywords in the manuscript file.

2. Author contributions should be indicated in the submission system.

3. Reference formatting: Please list up to 10 authors, followed by et al. for any additional authors.

4. Ensure the funding information listed in the manuscript and entered in the submission system is consistent. Currently, the EC | European Research Council (ERC) 616622 is missing from the manuscript file.

5. EV tables: Since Tables EV1 and EV2 are quite large, please convert them to EV Datasets. Update the source file names, titles, legends, and all manuscript callouts to reflect the new naming (i.e., Dataset EV1-EV2). Legends should be removed from the manuscript and included as a separate tab or sheet within each Excel file.

Additionally, Tables EV3 and EV4 should be renamed to Table EV1 and EV2, respectively. Please update the manuscript callouts accordingly, remove the legends from the manuscript, and include them directly above the tables within each Excel file.

6. Appendix figures should be listed individually in the Table of Contents, with the corresponding page number for each figure.

7. Add the missing callouts for Appendix Fig. S8.

8. Author Checklist: Please complete the general information box, and note that some sections in the last column are missing and need to be filled in.

9. Please address the following issues in figure legends:

- Please note that the exact p values are not provided in the legends of figures 3D, E

- Please note that the box plots need to be defined in terms of minima, maxima, centre, bounds of box and whiskers, and percentile in the legends of figures 3C, 4C

- Please note that information related to n is missing in the legend of figure 3C

10. Materials and Methods" should be renamed to "Methods".

Click on the link below to submit your revised paper.

As a matter of course, please make sure that you have correctly followed the instructions for authors as given on the submission

website.

Sincerely,
Jingyi

Jingyi Hou, PhD
Senior Editor
Molecular Systems Biology

***** PLEASE NOTE ***** As part of the EMBO Press transparent editorial process initiative (see our Editorial at <https://dx.doi.org/10.1038/msb.2010.72> , Molecular Systems Biology will publish online a Review Process File to accompany accepted manuscripts. When preparing your letter of response, please be aware that in the event of acceptance, your cover letter/point-by-point document will be included as part of this File, which will be available to the scientific community. More information about this initiative is available in our Instructions to Authors. If you have any questions about this initiative, please contact the editorial office (msb@embo.org).

Reviewer #1:

I am overall satisfied with the authors' responses. My further comments on each of my previous points are as follows.

First part

1.

Thank you for explaining the background. I find them to be reasonable.

2.

Yes I agree that PCA and clustering on Fig1F supported these claims.

3.

Thank you and this is good enough for me.

4/5

Okay

6

The explanation is acceptable to me. It is possible, however, to construct some kind of expectation, and put it as a dashed "baseline" in fig3c and 4c, so that we can conclude that (at least some of) these genes are, to some degree, adapted to the host tRNA pool? For example, randomly shuffling the correspondings between expression level and tRNA type ? I believe this would not change the authors conclusion.

7/8

These responses are fine.

9.

I respectfully disagree with some parts of the author's response. As I mentioned in my previous comment, I believe that aminoacyl-tRNA:EF-Tu:GTP is the actual supply of translation-ready tRNA. They may have used SOTA for determining tRNA abundance using the tRNA sequencing protocol they employed. However, after tRNA abundance, it must undergo aminoacylation, binding with EF-Tu, binding with GTP, codon-anticodon binding, and kinetic proof-reading. Thus, tRNA sequencing is not necessarily a reliable indicator of translational efficiency.

Having said that, I am also aware that there is currently no reliable way to determine the abundance of aminoacyl-tRNA:EF-Tu:GTP complex, and some studies indicate that the aminoacyl-tRNA fraction is uniform among different types of mature tRNA (PMID: 28586482). Hence, my comment is that, while tRNA sequencing is sufficient for this work already (as the focus is on mature tRNA pools), please discuss in greater detail the potential discrepancy between tRNA sequencing results and actual tRNA supply or translation efficiency. At least mention that changes in tRNA pool are not quantitatively equivalent to changes in translational efficiency.

Second part

I am satisfied with these responses and revisions.

In addition, I briefly read the comments of Reviewer #3. I would have hoped that some further validation would be carried out for 1b, such as qRT-PCR of the targeted genes. Nonetheless, I am comfortable with the current resolution of acknowledging the alternative possibilities, which is on lines 430-434 of page 15, not 423-425.

Reviewer #2:

Review of Aharon-Hefetz et al.

Summary

This study presents a timely, well-executed, and in-depth analysis of tRNA regulation and influence during HCMV infection in human foreskin fibroblasts. They employ DM-tRNA-seq to investigate both expression and modification dynamics during a time course of infection. They find several tRNAs that are differentially expressed during HCMV infection and more similar to tRNAs used during cellular differentiation, rather than proliferation. Additionally, tRNA differential expression in response to HCMV does not resemble that following interferon treatment, implying that virus modulation of tRNAs is distinct from the general interferon response. Differential tRNA regulation is not apparent for SARS-COV2 infection. Using mutation signatures in the DM-tRNA-seq data, they find minimal changes in tRNA modifications in response to HCMV, but note three specific tRNAs with reduced dihydrouridine modification and one specific tRNA with increased m3c modification.

To further elucidate the functional importance of specific tRNAs and their modification enzymes in cell growth, a tRNA-CRISPR screen was conducted. This screen utilized a cell competition assay, wherein the importance of various tRNA genes for cell growth was assessed based on the sgRNAs enriched or depleted following culture. The tRNA-CRISPR screen was also applied to HCMV-infected cells to identify which tRNAs and tRNA modifying enzymes are critical for viral infection. Surprisingly, there were few tRNA modification factor hits that contributed to growth or infection. By analyzing sgRNAs enriched in cells with low GFP expression, the authors identified two tRNA dependency factors and several tRNA restriction factors. The proviral tRNAs were further validated and shown to affect intracellular viral replication, as measured by vDNA copies. Attempts to reproduce knockouts of tRNA restriction factors outside of the screen did not replicate.

In summary, this work provides valuable insights into the specific tRNAs and tRNA modifying enzymes that modulate HCMV infection. Although the number of identified dependency factors was limited, these findings will inform and guide future research exploring the role of tRNAs during viral infection.

General Remarks

The authors were highly responsive to the previous round of review and this body of work now contributes a rigorous assessment of tRNA regulation during HCMV infection. Their data solidly supports their conclusions (except in one spot highlighted below). This work is timely, as takes recent descriptions of tRNA expression in response to herpesviruses towards a more functional analysis by applying tRNA-CRISPR to identify restriction and dependency tRNA and tRNA-related genes. The newly added follow-up experiments to validate an impact on HCMV replication increases the rigor and applicability of the work. This work will be of general interest in virology, as differential tRNA regulation is a widely applicable topic across virus families.

Major Point

Line 426-428: In light of the new data where tRNA restrictors did not validate, this language should include this caveat.

Minor Points

1. Cover art synopsis: misspelled knockout
2. Methods: Please insert methodology for vDNA extraction.
3. Lines 162-163: I am not sure the heat map specifically supports late stage regulation, rather that downregulation is present at most viral timepoints. Maybe change to: "...this reduction correlates with lower expression of these tRNAs during the course of infection (Fig 1F)."
4. Line 177: "these results indicate that HCMV infection selectively alters specific tRNA modifications" Given that the rest of the paper does not explore selective alterations in modification by HCMV (e.g. by comparison other viruses), the wording should be changed.
5. Figure 5: A western blot validating knockout for the protein hits and controls is required to make the results more convincing. If not, the authors should explain why they did not choose to validate protein KO by western blot. Is there not antibody available? The controls selected were not assessed to be either growth dependency or restrictive factors in the cited previous work (Hein & Weissman, 2022); thus, how were they selected?
6. Figure 5: To assist in the reader's understanding of what the values for "tRNA essentiality" mean, can you include text to the

left of the graph (i.e., "more essential" and "less essential" on the top/bottom of the graphs as appropriate).

7. Figure 6D: Authors should indicate if there was any statistical significance in these experiments or clarify why analysis is not appropriate.

8. Supplemental Fig 6B: Is the X-axis incorrect here? Shouldn't this be ancestor/GFP2?

9. Discussion: It might be worth commenting on the relationship between tRNA hits for growth versus infection. What is in common? What strikingly is not? Etc.

10. Note for authors: I am not asking for any additional analysis, just offering a suggestion/thought for the future. It might be possible that restriction factors were more difficult to reproduce because high MOIs can mask their effect. Perhaps you can try infecting at a low MOI and titrating infectious virus in the future to avoid this.

Att: Molecular Systems Biology Editor

Dear Dr. Jingyi Hou,

Re: "Essentiality and dynamic expression of the human tRNA pool during viral infection"

Manuscript Number: MSB-2024-12614R

Thank you for your message and for supporting the revised version of our manuscript for publication in *Molecular Systems Biology*. We appreciate the referee's response, enthusiasm, and good advice, and the editorial process.

In this revised submission, we have fully addressed all remaining minor issues raised by Reviewers #1 and #2. We have also completed all requested editorial updates. Specifically, we have added the missing information (keywords, callouts, p-values, etc.), corrected formatting inconsistencies (references, tables, appendix), and ensured that the funding information and author contributions are properly updated in both the manuscript and the submission system.

We hope that these revisions satisfactorily address all outstanding points, and that the manuscript can now proceed to formal acceptance.

Below, please find a detailed point-by-point response to the remaining reviewer comments.

Kind regards,
Yitzhak

Reviewer #1:

I am overall satisfied with the authors' responses. My further comments on each of my previous points are as follows.

We appreciate the reviewer's careful evaluation and are pleased that the revisions were found satisfactory. The two remaining comments have now been addressed as requested.

First part

1.

Thank you for explaining the background. I find them to be reasonable.

2.

Yes I agree that PCA and clustering on Fig1F supported these claims.

3.

Thank you and this is good enough for me.

4/5

Okay

6

The explanation is acceptable to me. It is possible, however, to construct some kind of expectation, and put it as a dashed "baseline" in fig3c and 4c, so that we can conclude that (at least some of) these genes are, to some degree, adapted to the host tRNA pool? For example, randomly shuffling the correspondings between expression level and tRNA type ? I believe this would not change the authors conclusion.

We thank the reviewer for suggesting this analysis, which is now included in the revised manuscript. Specifically, we generated randomized tAI values by randomly assigning tRNA expression profiles among tRNA types while preserving replicate structure. For each shuffled dataset, we calculated the tAI for every viral gene in each replicate and then averaged across replicates. This shuffling procedure was repeated 100 times, and the mean tAI across all genes for each round was recorded. We report the mean tAI across all randomization rounds.

This analysis was performed both for HCMV genes (based on tRNA expression profile of 24hpi samples) and SARS-CoV-2 genes (both uninfected and 6hpi). We added the results of the analyses to page 8, lines 208-210, 229-231, and Figs. 3C and 4C. We describe the analysis in the "Method" section, page 17, lines 511-515.

7/8

These responses are fine.

9.

I respectfully disagree with some parts of the author's response. As I mentioned in my previous comment, I believe that aminoacyl-tRNA:EF-Tu:GTP is the actual supply of translation-ready tRNA. They may have used SOTA for determining tRNA abundance using the tRNA sequencing protocol they employed. However, after tRNA abundance, it must undergo aminoacylation, binding with EF-Tu, binding with GTP, codon-anticodon binding, and kinetic proof-reading. Thus, tRNA sequencing is not necessarily a reliable indicator of translational efficiency.

Having said that, I am also aware that there is currently no reliable way to determine the abundance of aminoacyl-tRNA:EF-Tu:GTP complex, and some studies indicate that the aminoacyl-tRNA fraction is uniform among different types of mature tRNA (PMID:

28586482). Hence, my comment is that, while tRNA sequencing is sufficient for this work already (as the focus is on mature tRNA pools), please discuss in greater detail the potential discrepancy between tRNA sequencing results and actual tRNA supply or translation efficiency. At least mention that changes in tRNA pool are not quantitatively equivalent to changes in translational efficiency.

We fully agree with the reviewer's reservation that tRNA sequencing alone falls short of capturing the full picture of translational readiness, as aminoacylation, EF-Tu:GTP binding, and codon-anticodon interactions all contribute to the functional tRNA pool. We also appreciate the reviewer's acknowledgment of the technical challenges associated with directly measuring the aminoacyl-tRNA:EF-Tu:GTP complex. As suggested, we have now expanded the discussion section to explicitly address this limitation and to clarify that changes in tRNA abundance measured by sequencing are not necessarily quantitatively equivalent to changes in translational efficiency (page 14, lines 416–423).

Second part

I am satisfied with these responses and revisions.

In addition, I briefly read the comments of Reviewer #3. I would have hoped that some further validation would be carried out for 1b, such as qRT-PCR of the targeted genes. Nonetheless, I am comfortable with the current resolution of acknowledging the alternative possibilities, which is on lines 430-434 of page 15, not 423-425.

We thank the reviewer for carefully considering not only their own comments but also those of Reviewer #3. We appreciate the acknowledgment that our current approach, which includes discussing alternative possibilities, is satisfactory.

Reviewer #2:

Review of Aharon-Hefetz et al.

Summary

This study presents a timely, well-executed, and in-depth analysis of tRNA regulation and influence during HCMV infection in human foreskin fibroblasts. They employ DM-tRNA-seq to investigate both expression and modification dynamics during a time course of infection. They find several tRNAs that are differentially expressed during HCMV infection and more similar to tRNAs used during cellular differentiation, rather than proliferation. Additionally, tRNA differential expression in response to HCMV does not resemble that following interferon treatment, implying that virus modulation of tRNAs is distinct from the general interferon response. Differential tRNA regulation is not

apparent for SARS-COV2 infection. Using mutation signatures in the DM-tRNA-seq data, they find minimal changes in tRNA modifications in response to HCMV, but note three specific tRNAs with reduced dihydrouridine modification and one specific tRNA with increased m3c modification.

To further elucidate the functional importance of specific tRNAs and their modification enzymes in cell growth, a tRNA-CRISPR screen was conducted. This screen utilized a cell competition assay, wherein the importance of various tRNA genes for cell growth was assessed based on the sgRNAs enriched or depleted following culture. The tRNA-CRISPR screen was also applied to HCMV-infected cells to identify which tRNAs and tRNA modifying enzymes are critical for viral infection. Surprisingly, there were few tRNA modification factor hits that contributed to growth or infection. By analyzing sgRNAs enriched in cells with low GFP expression, the authors identified two tRNA dependency factors and several tRNA restriction factors. The proviral tRNAs were further validated and shown to affect intracellular viral replication, as measured by vDNA copies. Attempts to reproduce knockouts of tRNA restriction factors outside of the screen did not replicate.

In summary, this work provides valuable insights into the specific tRNAs and tRNA modifying enzymes that modulate HCMV infection. Although the number of identified dependency factors was limited, these findings will inform and guide future research exploring the role of tRNAs during viral infection.

General Remarks

The authors were highly responsive to the previous round of review and this body of work now contributes a rigorous assessment of tRNA regulation during HCMV infection. Their data solidly supports their conclusions (except in one spot highlighted below). This work is timely, as takes recent descriptions of tRNA expression in response to herpesviruses towards a more functional analysis by applying tRNA-CRISPR to identify restriction and dependency tRNA and tRNA-related genes. The newly added follow-up experiments to validate an impact on HCMV replication increases the rigor and applicability of the work. This work will be of general interest in virology, as differential tRNA regulation is a widely applicable topic across virus families.

We thank the reviewer for their thoughtful and constructive feedback throughout the review process, and we greatly appreciate the positive comments that highlight the strengths of our study and the revisions. We are pleased that the reviewer found the revised manuscript well-improved. The remaining comments have now been fully addressed as requested.

Major Point

Line 426-428: In light of the new data where tRNA restrictors did not validate, this language should include this caveat.

We now clearly state this caveat in both the Results section (Page 12, Line 334) and in the Discussion (Page 15, Lines 438-440).

Minor Points

1. Cover art synopsis: misspelled knockout

We thank the reviewer for noting the misspelling. We have now corrected it.

2. Methods: Please insert methodology for vDNA extraction.

We inserted the requested methodology (page 21, line 627).

3. Lines 162-163: I am not sure the heat map specifically supports late stage regulation, rather that downregulation is present at most viral timepoints. Maybe change to: "...this reduction correlates with lower expression of these tRNAs during the course of infection (Fig 1F)."

We agree with the reviewer; thus, we have changed the phrasing accordingly (page 6, lines 161-162).

4. Line 177: "these results indicate that HCMV infection selectively alters specific tRNA modifications" Given that the rest of the paper does not explore selective alterations in modification by HCMV (e.g. by comparison other viruses), the wording should be changed.

The reviewer's point is well taken; we have changed the phrasing accordingly (page 7, lines 176-177).

5. Figure 5: A western blot validating knockout for the protein hits and controls is required to make the results more convincing. If not, the authors should explain why they did not choose to validate protein KO by western blot. Is there not antibody available? The controls selected were not assessed to be either growth dependency or restrictive factors in the cited previous work (Hein & Weissman, 2022); thus, how were they selected?

We agree that knockout validation could be made to provide additional support for the results.

The screen included 4 sgRNAs per target, which were taken from a well-established and widely validated CRISPR knockout library ([10.1038/s41467-018-07901-8](https://doi.org/10.1038/s41467-018-07901-8)), that has been extensively benchmarked in previous studies ([10.1016/j.cell.2018.10.024](https://doi.org/10.1016/j.cell.2018.10.024), [10.1186/s12943-024-01987-z](https://doi.org/10.1186/s12943-024-01987-z)). It is important to note that the knockout efficiency in the screen was validated functionally, as the control gene knockouts produced the expected effects, confirming the overall validity and robustness of the screening approach. Therefore, additional validation of knockout efficiency was deemed unnecessary. The control hits, both for cell growth and HCMV infection, were chosen based on the Hein and Weismann's results. We used this paper as a reference for both cell type (HFF) and viral infection (HCMV Merlin strain). We looked for genes whose knockdown/knockout affected one phenotype and not the other. The effect of the chosen gene hits from Hein's paper is shown in Figure S7G. This was now better explained on page 9, lines 256-258.

6. Figure 5: To assist in the reader's understanding of what the values for "tRNA essentiality" mean, can you include text to the left of the graph (i.e., "more essential" and "less essential" on the top/bottom of the graphs as appropriate).

We added an arrow indicating higher essentiality. We also described it in the legend (page 34, lines 1073, 1078).

7. Figure 6D: Authors should indicate if there was any statistical significance in these experiments or clarify why analysis is not appropriate.

We thank the reviewer for pointing out the lack of statistical information. We have now performed a t-test with correction for multiple comparisons, and the differences observed in Figure 6D were found to be statistically significant. These results have been added to the figure and legend (page 35, lines 1096–1098).

8. Supplemental Fig 6B: Is the X-axis incorrect here? Shouldn't this be ancestor/GFP2?

Supplemental Fig. 6B compares a sample group of lowly infected cells (GFP2) to uninfected cells (ancestor), as was asked by the reviewer in the first revision. As is typically done in sample comparisons, we calculated the log₂ fold change (log₂FC) of gene hits between the treated sample and the control, i.e., GFP2/ancestor.

9. Discussion: It might be worth commenting on the relationship between tRNA hits for growth versus infection. What is in common? What strikingly is not? Etc.

Indeed, this interesting comparison was missed in the discussion. We have now added a new discussion item comparing the two screens (page 15, lines 443-450).

10. Note for authors: I am not asking for any additional analysis, just offering a suggestion/thought for the future. It might possible that restriction factors were more difficult to reproduce because high MOIs can mask their effect. Perhaps you can try infecting at a low MOI and titering infectious virus in the future to avoid this.

We thank the reviewer for this suggestion and fully agree. Originally, we aimed to reproduce the same conditions as those in which we performed the CRISPR screen, also in the validation experiments. The results of the restriction factors made us reach the same conclusion as suggested by the referee. Thus, we plan to apply the proposed low-MOI infection and virus titration approach in future experiments.

8th Dec 2025

Manuscript number: MSB-2024-12614RR

Title: Essentiality and dynamic expression of the human tRNA pool during viral infection

Dear Dr. Pilpel,

Thank you again for sending us your revised manuscript. We are now satisfied with the modifications made and I am pleased to inform you that your paper has been accepted for publication.

You may qualify for financial assistance for your publication charges - either via a Springer Nature fully open access agreement or an EMBO initiative. Check your eligibility: <https://link.springer.com/journal/44320/how-to-publish-with-us>

Sincerely,
Jingyi

Jingyi Hou, PhD
Senior Editor
Molecular Systems Biology

>>> Please note that it is Molecular Systems Biology policy for the transcript of the editorial process (containing referee reports and your response letter) to be published as an online supplement to each paper. If you do NOT want this, you will need to inform the Editorial Office via email immediately. More information is available here: <https://link.springer.com/partners/embo-press/editorial-policies#Peer%20review>

Rev_Com_number: RC-2024-02487

New_manu_number: MSB-2024-12614RR

Corr_author: Pilpel

Title: Essentiality and dynamic expression of the human tRNA pool during viral infection